# Tuning instability of non-columnar neurons in the salt-and-pepper whisker map in somatosensory cortex

Han Chin Wang[1], Amy M. LeMessurier[1,2] & Daniel E. Feldman [1] ✉

Rodent sensory cortex contains salt-and-pepper maps of sensory features, whose structure is not fully known. Here we investigated the structure of the salt-and-pepper whisker somatotopic map among L2/3 pyramidal neurons in somatosensory cortex, in awake mice performing one-vs-all whisker discrimination. Neurons tuned for columnar (CW) and non-columnar (non-CW) whiskers were spatially intermixed, with co-tuned neurons forming local (20 μm) clusters. Whisker tuning was markedly unstable in expert mice, with 35-46% of pyramidal cells significantly shifting tuning over 5-18 days. Tuning instability was highly concentrated in non-CW tuned neurons, and thus was structured in the map. Instability of non-CW neurons was unchanged during chronic whisker paralysis and when mice discriminated individual whiskers, suggesting it is an inherent feature. Thus, L2/3 combines two distinct components: a stable columnar framework of CW-tuned cells that may promote spatial perceptual stability, plus an intermixed, non-columnar surround with highly unstable tuning.

Rodent sensory cortex contains highly intermixed representations of sensory features, unlike columnar maps in primates and carnivores[1]. This salt-and-pepper intermixing of differently tuned neurons is particularly strong in layer (L) 2/3, the major ascending output layer. While salt-and-pepper maps appear non-topographic or poorly topographic, they may contain hidden structure relevant for sensory coding or plasticity. To test this, we studied mouse whisker somatosensory cortex (S1), where L2/3 contains spatially intermixed neurons tuned for different whiskers, as shown by 2-photon imaging[2–6] and single-unit recording[7,8], and anatomical column boundaries are marked by barrels in L4[9]. Unlike L2/3, L4 has precise columnar somatotopy, suggesting that salt-and-pepper organization in L2/3 is constructed by cortical circuits rather than inherited from L4[10]. We searched for structure in the L2/3 salt-and-pepper map by examining its fine-scale spatial organization and its stability over time in awake, whisker-attentive mice.

Precise whisker receptive fields and somatotopic organization in S1 are currently known only from anesthetized or sedated mice. Detailed map structure in awake mice is unclear, because experiments

have mapped responses to just 1 or 2 whiskers, rather than a full set of local whiskers. Here, we applied calibrated deflections of 9 whiskers as whisker-attentive mice performed a passive, one-vs-all whisker discrimination task. Using 2-photon imaging, we measured whisker receptive fields from L2/3 pyramidal (PYR) neurons in task expert mice. L2/3 exhibited robust salt-and-pepper somatotopy in these highly trained mice, with local clustering of similarly tuned neurons on a 20 μm scale.

We also probed map structure over time. Single-neuron feature selectivity and responsiveness can be remarkably unstable across days in some cortical areas in adults, even outside of active learning[11,12]. Such representational drift may require compensation by downstream areas to stably read changing population codes[12–14], or may be constrained to non-coding dimensions to minimize its impact on information representation[12,15–17]. Representational drift is common in higher cortical areas that synthesize sparse, non-topographic population codes through learning[17–19], but is thought to be minimal for local sensory feature coding in primary sensory cortex, where topographic

[1]Department of Molecular & Cell Biology, and Helen Wills Neuroscience Institute, University of California Berkeley, Berkeley, CA 94720, USA. [2]Present address: Neuroscience Institute and Department of Neuroscience and Physiology, New York University School of Medicine, New York, NY 10016, USA. ✉e-mail: dfeldman@berkeley.edu

representations predominate[16,20–28]. However, several studies report tuning instability for a minority of neurons (~20%) in primary sensory cortex[16,22,23,25,27,29,30]. We hypothesized that within salt-and-pepper maps in primary sensory cortex, unstable, non-topographic subnetworks may be intermixed with stable, more topographically organized subnetworks, with distinct roles in representational stability, flexibility, and learning[19,22,25].

To test this hypothesis, we performed longitudinal 2-photon Ca²⁺ imaging in expert mice with consistent task performance. Results showed that that the whisker map is markedly unstable, with ~40% of PYR cells significantly changing whisker tuning over a 5–18-day period. This instability was structured within the whisker map, in a way which reveals that L2/3 is not simply a poorly topographic map, but instead combines two distinct components: a topographically accurate columnar representation of each whisker with highly stable tuning, plus a non-columnar surround with highly dynamic, unstable tuning. This separation of stable and unstable networks may explain how stability and flexibility are balanced in salt-and-pepper sensory maps, and how downstream areas are able to read an unstable, shifting neural code.

## Results

### One-versus-all whisker discrimination task for awake receptive field measurements

We developed a head-fixed whisker discrimination task to allow measurement of receptive fields in awake, whisker-attentive mice (Fig. 1a). Mice ($n = 10$) had 9 whiskers inserted in a $3 \times 3$ piezoelectric actuator array, and on randomly interleaved trials were presented with either an all-whisker stimulus, one of 9 single-whisker stimuli, two different tone stimuli, or no-stimulus blanks. Whisker stimuli were 0.5-s deflection trains. Only the all-whisker stimulus was rewarded (S+). Mice learned to lick to the all-whisker S+, but not to the single-whisker or other stimuli, which were unrewarded (S–). Expert performance, defined as hit rate >80% and false alarm rate <25%, was achieved $11.2 \pm 1.18$ days after the introduction of all S– stimuli. In expert mice, false alarm licking was more common to single-whisker S– than sound S– or blanks, indicating that mice attended whisker stimuli more than auditory distractors (Fig. 1b, c). This task design allows single-whisker stimuli to be used to map receptive fields in whisker-attentive mice without lick contamination or a delay period. Whisker movements were minimal, occurring on less than 5% of whisker S– trials (Supplementary Fig. 1a, b). Paralysis of active whisking by botulinum toxin (Botox) injection in the whisker pad did not affect task performance ($n = 4$ mice, see below) or qualitatively alter whisker-evoked activity in S1 (Supplementary Fig. 2a–d). Thus, this is a passive one-versus-all whisker discrimination task.

To image the activity of L2/3 PYR neurons, GCaMP6s[31] was expressed either virally in Emx1-Cre mice or transgenically in Drd3-Cre:TIGRE2.0-GCaMP6s (Ai162) mice[32]. We imaged neural activity during behavior using 2-photon Ca²⁺ imaging (Fig. 1d). Whisker responses and receptive fields were measured from the single-whisker S– trials (Fig. 1e). False-alarm trials were excluded to avoid any lick-related activity and motion artifacts. Imaged neurons were localized post-hoc relative to anatomical column boundaries (Fig. 1d and Supplementary Fig. 1d) by reconstructing each imaging field relative to cytochrome oxidase-stained barrels in L4[10,33]. All imaging was performed in expert mice after task learning.

### L2/3 PYR neurons show locally heterogeneous tuning overlaid on a global whisker map

Whisker receptive fields have not been measured in awake, whisker-attentive mice. We analyzed 7393 L2/3 PYR cells from 40 imaging fields in 6 mice with viral GCaMP6s expression and 4 mice with transgenic GCaMP6s expression. 30.1% (2228/7393) of PYR cells were significantly whisker-responsive, consistent with sparse coding in L2/3 of S1[34–37].

Whisker tuning was generally narrow, and for each cell only a subset of whiskers drove significant $\Delta F/F$ responses (Fig. 1d, e). Of responsive cells, 76% had a single best whisker (BW) that elicited a statistically stronger response than any other whisker. The remaining cells had several statistically equivalent best whiskers (eBWs, usually 2–3), and thus were broadly tuned.

Cells tuned for different whiskers were intermixed in L2/3 of each column (e.g., Fig. 1d), as in anesthetized mice[2,3,10]. We classified cells within whisker column boundaries as being CW-tuned or non-CW tuned. The latter was defined conservatively as cells that responded to a non-CW significantly more than the CW, by permutation test. CW- and non-CW tuned neurons represent heavy tails of a tuning distribution quantified by CW tuning dominance index (Fig. 1e). Both CW- and non-CW tuned cells had reliable tuning and similar whisker-evoked response strength (Fig. 1e, f and Supplementary Fig. 1e, f). Both were sharply tuned, but tuning was slightly sharper for CW-tuned cells than non-CW tuned cells or cells located over L4 septa (Fig. 1f, g). CW-tuned neurons responded on a higher fraction of trials (Fig. 1h). To ensure that tuning measurements were not biased by any small whisker movements, we compared sessions in Drd3-Cre:TIGRE2.0 mice before and after Botox injection, and found no difference in mean tuning properties (Supplementary Fig. 2).

We quantified the salt-and-pepper intermixing of PYR neurons tuned to different whiskers (example field: Fig. 1d; full population: Fig. 2a). On average, 52% of responsive PYR neurons in each column were tuned to the CW, and the rest were tuned to a non-CW (Fig. 2b). Correspondingly, the set of PYR neurons tuned to a given whisker, termed the tuning ensemble, spanned several columns, with only 42% of cells located in the anatomical column for their BW (Fig. 2c, d). Each tuning ensemble was centered on its topographically appropriate column (Fig. 2e). Within each column, a rough subcolumnar map of whisker tuning was evident (Supplementary Fig. 3a). Thus, salt-and-pepper tuning heterogeneity is overlaid on the classical somatotopic whisker map, both across and within columns. Within each tuning ensemble, tuning sharpness and response magnitude to the BW gradually decreased with cortical distance from the BW column center (Fig. 2f). Across all whisker-responsive cells, tuning preference to a given whisker, and response magnitude for that whisker, decreased with distance from the column center, with the greatest decrease occurring in the near half of adjacent columns (Fig. 2g). This indicates a gradual tuning gradient in L2/3 rather than sharp boundaries at column edges[2,10]. Map structure was similar in viral Emx1-Cre and transgenic Drd3-Cre mice, which were combined for the above analysis (Supplementary Fig. 3b, c and Supplementary Fig. 4). Abolishing active whisking with Botox did not change whisker map organization (Supplementary Fig. 2e, f).

Thus, the L2/3 whisker map in awake, whisker-attentive mice consists of a topographic, columnar core of CW-tuned neurons in each anatomical column, intermixed with an equal number of neurons tuned for nearby non-CW whiskers. The tuning ensemble for each whisker is distributed and includes non-CW-tuned neurons that form a diffuse surround spilling broadly outside of the whisker's anatomical column. This is similar to anesthetized mice[2–4,10].

### Tuning clusters within the salt-and-pepper map

Is the salt-and-pepper map purely random at the local level, or are cells locally clustered by their tuning preference, like 'mini-columns' in rodent V1[38–41]? To test for this, we examined signal correlations (tuning similarity) between co-columnar pairs of whisker-responsive neurons. Signal correlation fell off with distance between neurons (Fig. 3a, red line), with a linear relationship between 30 and 200 μm (dashed line) that reflects the subcolumnar tuning gradient. Neurons <20 μm apart had signal correlations that were significantly higher than the extrapolated linear regression, suggesting local tuning clusters. Such local signal correlations exist throughout the column and not just at its

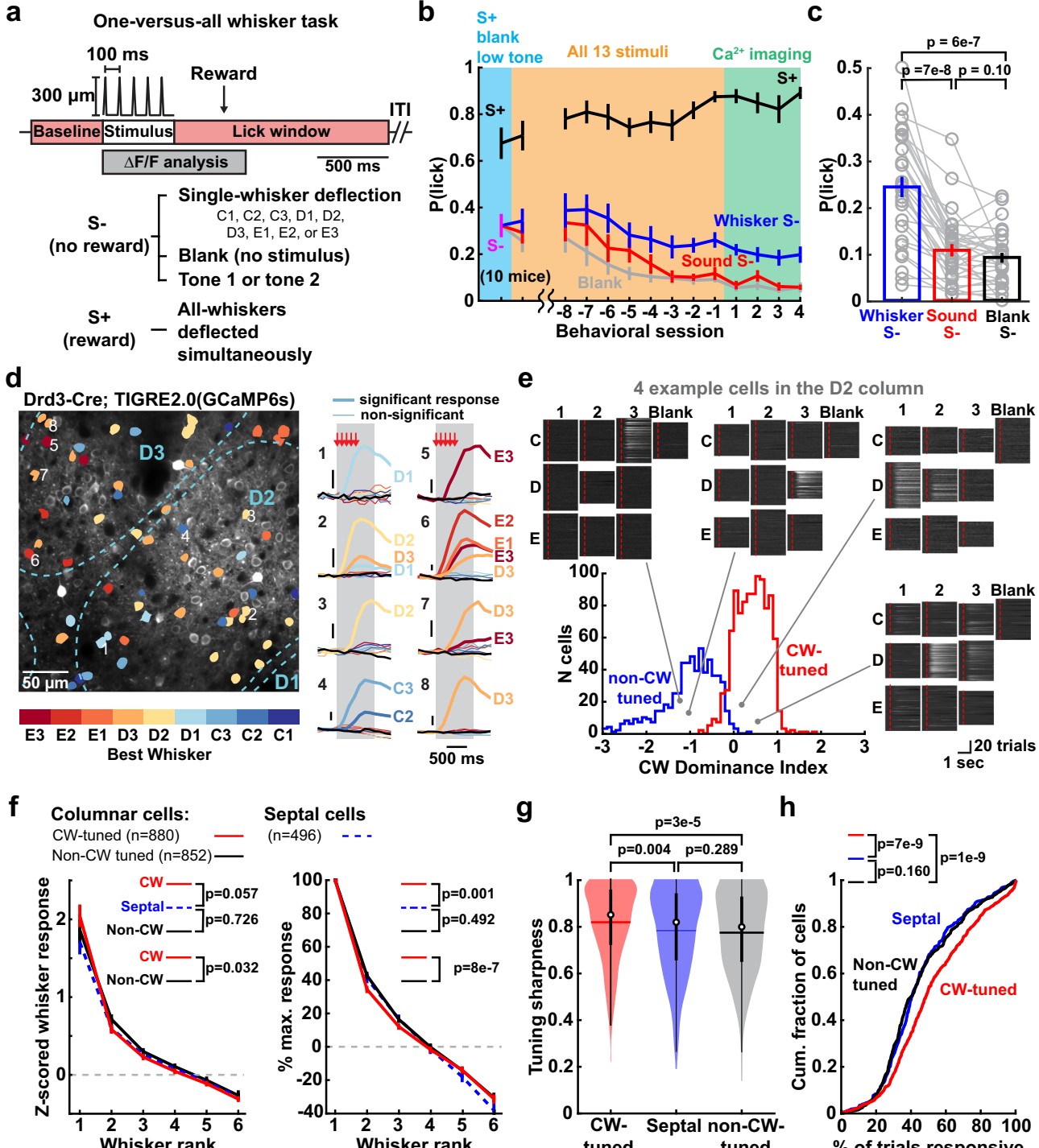

**Fig. 1 | Whisker tuning of L2/3 PYR neurons in awake, whisker-attentive mice.**
**a** Trial structure and stimuli for the one-versus-all whisker task. **b** Behavioral per-formance after training on a partial stimulus set (blue), during training with all S+ and S− stimuli (yellow, showing the first session and the last 8 sessions prior to imaging), and once mice are experts (green, showing the first 4 imaging sessions). **c** Lick probability for each S− stimulus type during days −3 to −1 in (**b**). Each line is one session in one mouse. Statistics: Friedman test, two-sided. **d** Example imaging field showing barrel boundaries and GCaMP6s-expressing PYR cells color-coded for BW identity. Right: mean whisker-evoked ΔF/F traces for 8 cells. Thick traces and whisker labels show significant responses; thin traces are non-significant responses; black traces are blanks. Arrows, whisker deflection. Gray, response analysis window. **e** Distribution of columnar whisker dominance index (CWDI) for cells located within a column (i.e., non-septal cells). CWDI quantifies a cell's relative response to

the columnar vs. its strongest non-columnar whisker (see "Methods"). Single-trial ΔF/F traces are shown for each single-whisker trial and blank trial, for 4 cells imaged in the D2 column. Dash, stimulus onset. ΔF/F traces are normalized to maximum for that cell. **f** Mean rank-ordered whisker tuning curves for CW-tuned neurons, non-CW tuned neurons, and septal-related neurons. All cells with the BW and at least 5 adjacent whiskers in the piezo array were included. Responses are normalized to activity in blank trials (left) or to the strongest whisker response (right). *p*-values are for cell type factor in unbalanced two-way ANOVA. **g** Tuning sharpness for responsive neurons. Circles are medians, and horizontal lines are means. Thick vertical lines are interquartile range and thin vertical lines are 1.5 × interquartile range. Statistics: rank-sum, two-sided. **h** Fractions of individual trials with sig-nificant whisker response, for each cell type. Statistics: KS, two-sided. All error bars are SEM.

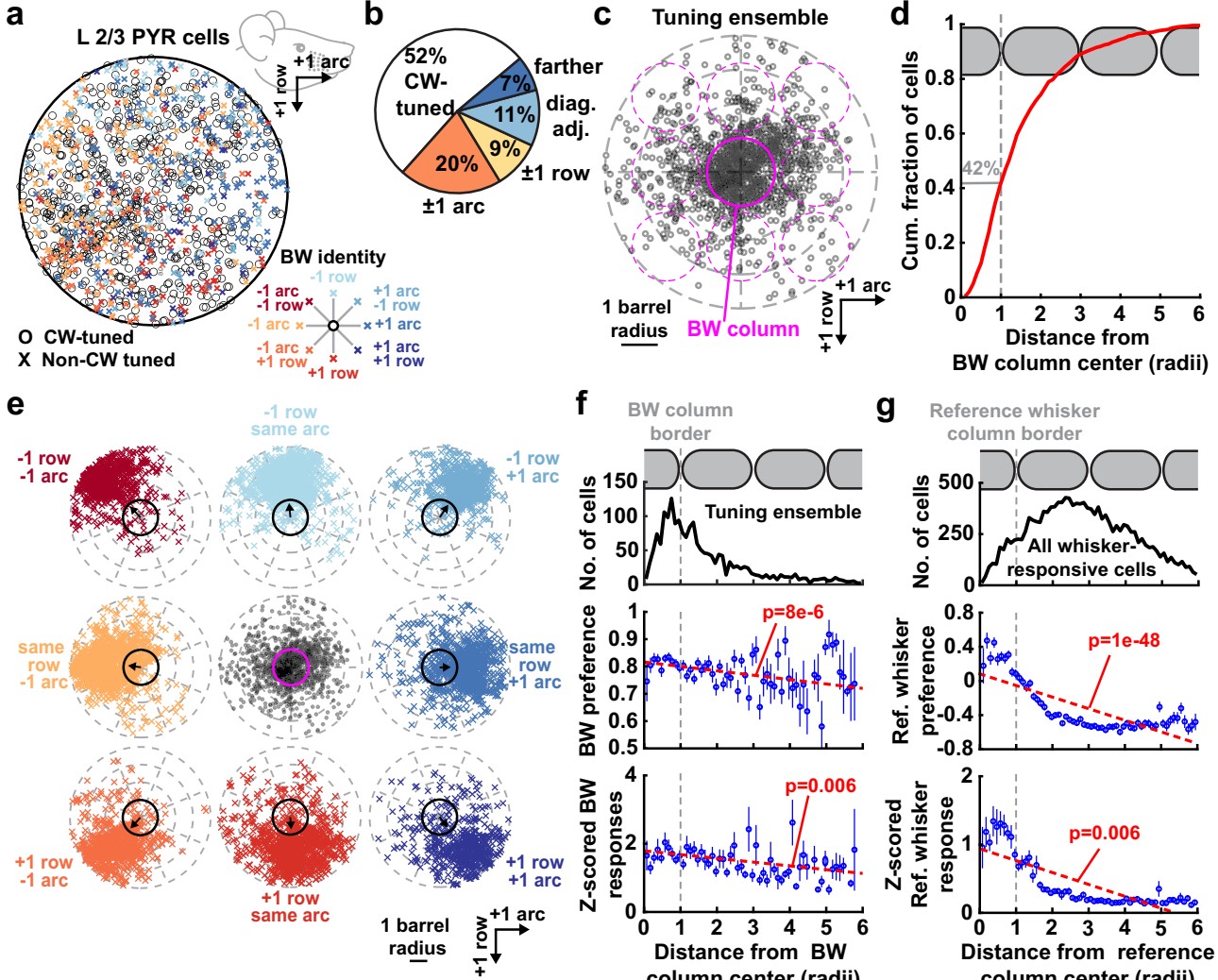

**Fig. 2 | Whisker map topography for L2/3 PYR neurons in one-versus-all whisker-cued mice. a** Whisker tuning (BW identity) and location for all responsive PYR neurons located within column boundaries ($n = 1732$). Circles, CW-tuned neurons. X's, non-CW tuned neurons. Colors show BW identity relative to the column where the cell resided. **b** Fraction of columnar cells tuned to different BWs. **c** Spatial distribution of the tuning ensemble, shown by plotting the location of each PYR neuron relative to its BW column. This includes both columnar and septal cells. Dashed magenta circles are nearby columns. **d** Distance from BW column center for cells in (**c**). **e** Locations of cells tuned to each of the 8 whiskers surrounding a central reference whisker. Locations are plotted relative to the reference whisker column (black or magenta circle). Arrows show circular mean of cell locations. **f** BW tuning preference and mean BW response magnitude for all

whisker-responsive neurons, as a function of the cell's distance from its BW column center. This quantifies the tuning and response gradient within a whisker's tuning ensemble. Dashed lines, the linear regression. Statistics: two-sided $t$ test for non-zero slope. $n = 2228$ cells. Error bars: SEM. **g** Average tuning preference and response magnitude for a reference whisker for all whisker-responsive neurons, as a function of cell distance from the reference whisker's anatomical column center. This measurement was performed for each whisker in the piezo array; thus each cell ($n = 2228$) is represented 9 times. This quantifies the response gradient to a given whisker across all responsive cells no matter their tuning, and thus represents the point representation of that whisker in L2/3. Dashed lines, the linear regression. Statistics: two-sided $t$ test for nonzero slope. Error bars: SEM.

center (Fig. 3b). Local clustering was also observed for two other measures of tuning similarity—the difference in tuning center of mass (Fig. 3c) and the probability that two cells share the same BW (Supplementary Fig. 5a). There was no clustering for whisker response magnitude (Fig. 3d). Restricting analysis to the D2 column, where sampling of surround whiskers was most complete, revealed similar clustering (Supplementary Fig. 5b–d).

To reduce neuropil contamination as an artifactual source of local tuning similarity, fluorescence was extracted in our entire study using the CaImAn algorithm which robustly removes signal contamination between neighboring neurons[42]. Moreover, we performed additional imaging in 2 mice expressing nucleus-targeted H2B-GCaMP6s[43]. In these mice, GCaMP6s is excluded from the cytosol, eliminating neuropil contamination, and whisker tuning can still be measured (Fig. 3e).

Tuning clusters were also observed using nuclear GCaMP6s, confirming that these are not due to neuropil contamination (Fig. 3f–h and Supplementary Fig. 5e, f). Thus, L2/3 contains local tuning clusters superimposed on the mean subcolumnar tuning gradient.

## Map differences between whisker-attentive and sound-attentive mice

To test if behavioral training on the one-versus-all whisker task shaped map organization, we compared the 6 viral Emx1-Cre mice performing this task (termed the whisker-cued task) with 5 additional viral Emx1-Cre mice that were presented with the identical stimulus set, but were trained to lick to one of the two tones (S+), but not to the other tone (S−) or to whisker stimuli (Fig. 4a). This is termed the sound-cued task. Sound-cued mice exhibited higher false-alarm licking to the tone

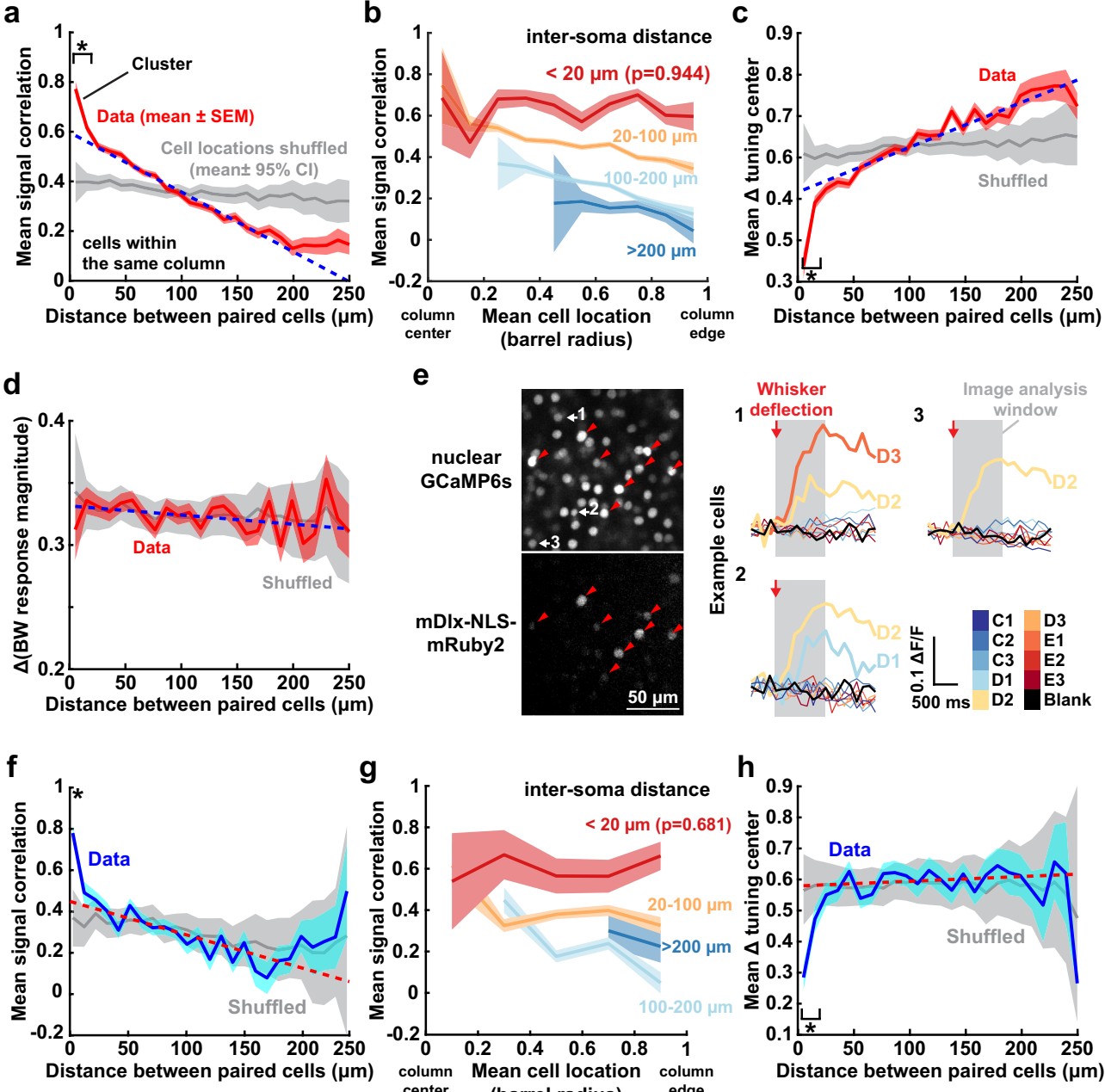

**Fig. 3 | Existence of local tuning clusters within a whisker column. a** Mean signal correlation between pairs of co-columnar neurons as a function of distance between neurons (10 μm bins). Red: measured data. Gray: spatially shuffled neurons. Dashed line: linear regression of measured data from 30 to 200 μm. *p*-value for significant difference from extrapolated linear regression: 1e−6, 0.001. **b** Mean signal correlation for co-columnar neuron pairs, separated by inter-soma distance, as a function of mean radial distance of the pair from the column center. Statistics: two-sided t-test for nonzero slope. Pairs <20 μm apart were more similarly tuned no matter where they were located in the column. **c** Mean difference in tuning center of mass between neuron pairs, as a function of separation between cells. Plotted as in (**a**). *p*-value for significant difference from extrapolated linear regression: 3e−4, 0.036. **d** Difference in BW response magnitude of co-columnar PYR neurons, as a function of distance between neuron pairs. Plotted as in (**a**). **e** Example field showing nucleus-localized GCaMP6s expressed pan-neuronally (upper panel), with interneurons labeled with mRuby2 via AAV1-mDlx-NLS-mRuby2 (arrowheads). Right: Mean Δ*F*/*F* traces for 3 example mRuby2-negative (putative PYR) cells from the field on the left. Conventions as in Fig. 1d. **f** Same as (**a**), but using data from putative PYR cells in the nuclear GCaMP6s experiment. Blue: observed data. *p*-value for significant difference from extrapolated linear regression: 6e−5. **g** Same as (**b**), but using data from nucleus-localized GCaMP6s. Statistics: two-sided *t* test for nonzero slope **h**. Same as (**c**), but using data from nuclear GCaMP6s. *p*-value for significant difference from extrapolated linear regression: 5e−4, 0.037. All shading is SEM except shuffled data in (**a**), (**c**), (**d**), (**f**), (**h**), which is 95% CI. All asterisks indicated significant difference by one-sided permutation test within each bin and corrected for multiple comparisons with false discovery rate 0.05 (see "Methods").

S− than to whisker stimuli, indicating that they attended to auditory stimuli (Fig. 4b, c). PYR cell response amplitude and tuning sharpness were not different between sound-cued and whisker-cued mice (Fig. 4d–h). Sound-cued mice also exhibited a salt-and-pepper map, but showed slightly less tuning heterogeneity, evidenced by a spatially narrower tuning ensemble and an increased fraction of PYR cells that were tuned for the CW (Fig. 4i, j). These effects were confirmed after subsampling to ensure similar spatial distributions of neurons between these two conditions (Supplementary Fig. 6). Thus, maps in whisker-cued and sound-cued mice are grossly similar, indicating the map

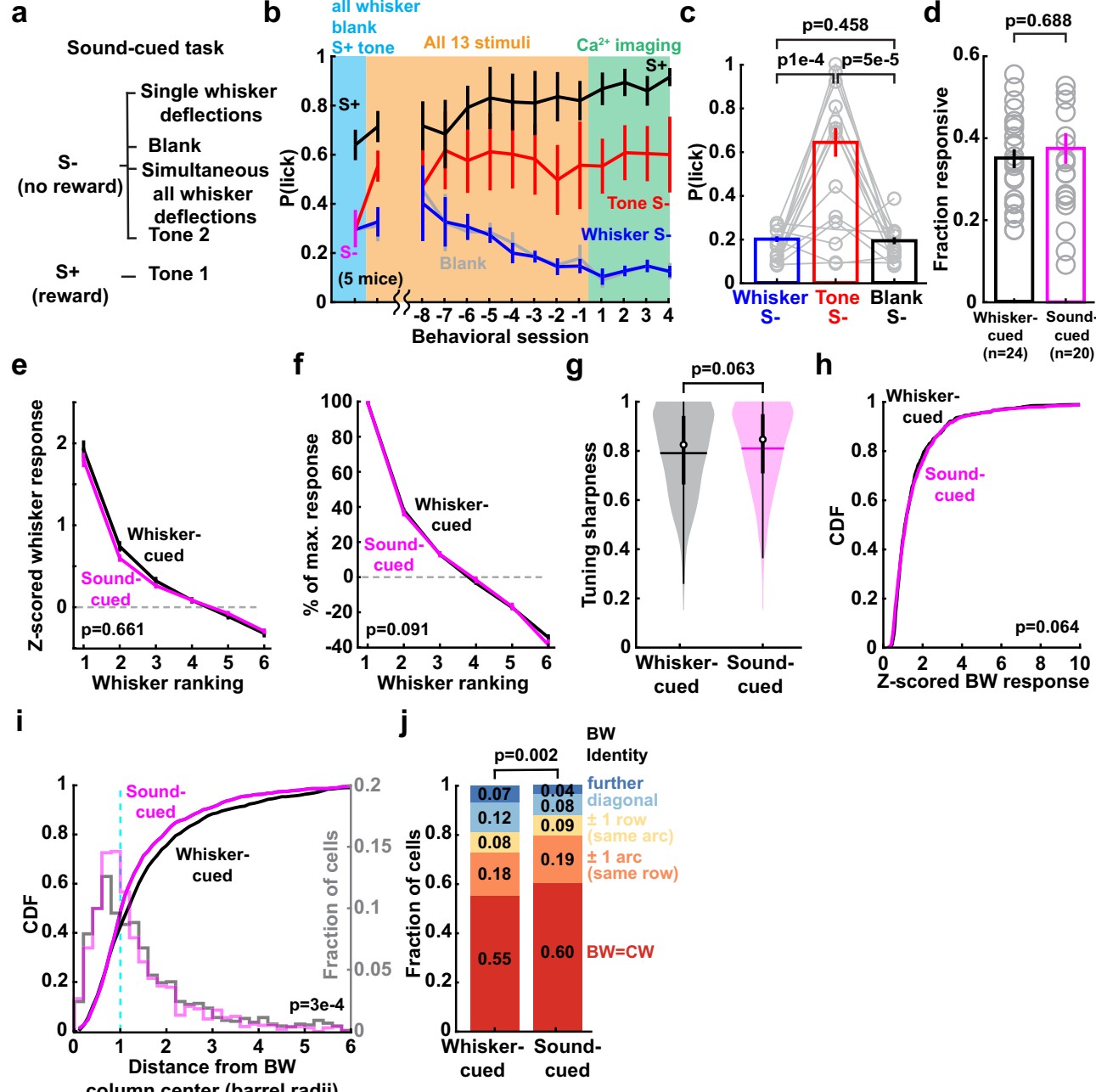

**Fig. 4 | Whisker map differences between whisker-cued and sound-cued mice.**
**a** Stimulus-reward assignment for sound-cued task. **b** Behavioral performance during training on the sound-cued task, including after training on a partial stimulus set (blue), during training with all S+ and S− stimuli (yellow, showing the first session and the last 8 sessions prior to imaging), and once mice are experts (green, showing the first 4 imaging sessions). **c** Lick probability for each S− stimulus type during days −3 to −1 in (**b**). Each gray line is one session in one mouse. $n = 5$ mice. Statistics: Friedman test, two-sided. **d** Fraction of whisker-responsive neurons in each imaging field for one-versus-all (whisker-cued) mice and sound-cued mice. Each circle is one imaging field. Statistics: rank-sum, two-sided. **e,f** Mean rank-ordered whisker tuning curves for cells in whisker-cued and sound-cued mice. Plotted as in Fig. 1f. *p*-values are for cue type factor in unbalanced two-way ANOVA. **g** Mean tuning sharpness of responsive neurons. Statistics: rank-sum, two sided. **h** BW response magnitude for cells in whisker-cued and sound-cued mice. Statistics: KS, two-sided. **i** Cumulative and binned distributions of distance from responsive cells to their BW column center. Statistics: KS, two sided. **j** Identity of BW for all responsive cells in a column. Statistics: $\chi^2$, two-sided. All error bars or shadings are SEM. (**e**–**j**): $n = 1387$ cells in whisker-cued mice and $n = 1217$ cells in sound-cued mice.

structure we characterized above was not the product of the specific training paradigm.

## Pronounced tuning instability revealed by longitudinal Ca²⁺ imaging

We tested for additional structure by examining whisker tuning over time. Simple sensory feature tuning in topographic sensory areas is generally thought to be stable outside of active learning[22,24,25,30,37]. In S1,

single-neuron tuning for whisker direction and touch features, as well as overall responsiveness, have been reported to be highly stable over days[22,24,37], but the stability of somatotopic tuning is unclear[22]. Because of its intermixed whisker topography, we hypothesized that L2/3 may exhibit the tuning drift that characterizes many high-order, non-topographic cortical areas[17-19,44,45]. Moreover, tuning instability may be organized within L2/3 in a way that reveals how flexibility and stability are balanced in cortical codes.

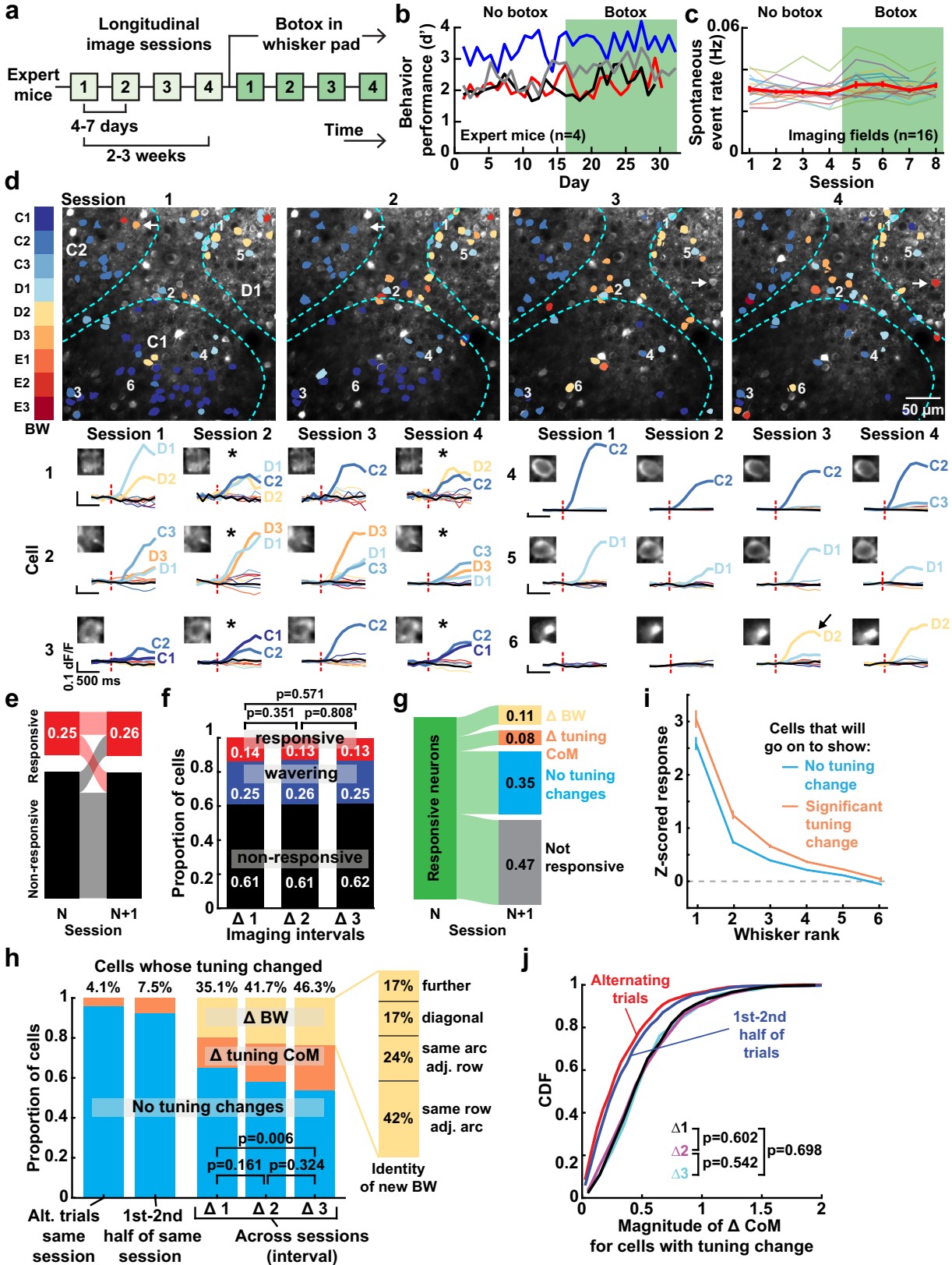

We assessed tuning stability in mice expert on the one-versus-all whisker task, using Drd3-Cre:TIGRE2.0 mice in which GCaMP6s expression is stable over months. We longitudinally imaged the same PYR neurons across 4 sessions spaced 4–7 days apart. We then injected Botox to immobilize the whiskers and conducted another 4 imaging sessions from the same neurons (Fig. 5a). Mice were expert in the task

before imaging started, and maintained stable performance across all sessions (Fig. 5b). The rate of spontaneous GCaMP6s events remained stable across sessions, suggesting stable GCaMP expression (Fig. 5c).

We first characterized tuning stability in the 4 pre-Botox sessions. Across 4 mice, 4204 neurons were imaged in at least 2 of 4 sessions, and 2771 neurons in all 4 sessions. A large number of PYR neurons

**Fig. 5 | Receptive field dynamics measured during longitudinal Ca²⁺ imaging.**
**a** Timeline of longitudinal imaging sessions from one imaging field. **b** Behavioral performance over the period of longitudinal imaging. Each line is one mouse. **c** Mean spontaneous event rate for each imaging field across sessions. Thick line, grand mean ± SEM across fields. **d** Example field imaged for 4 sessions (total 2–3 weeks), with responsive cells color-coded by their BW on each day. Mean $\Delta F/F$ traces to deflection of individual whiskers are shown for 6 selected cells on each session (inset: cell images). Thick traces with whisker labels are significant responses. Dash, stimulus onset. Asterisks indicate tuning that is changed from prior session. Arrow indicates a shift from non-responsive to responsive. **e** Changes in responsiveness for cells tracked longitudinally across a Δ1 session interval. Numbers are fraction of cells. **f** Fraction of neurons which were stably non-responsive, stably responsive, or wavering between responsive and non-responsive over Δ1, Δ2, or Δ3 session intervals. Statistics: $\chi^2$, two-sided. **g** Changes in tuning observed across a Δ1 session interval, for neurons which were responsive in the first session. **h** Proportion of stably responsive cells whose tuning significantly changed within an imaging session (left two bars) and over Δ1, Δ2, or Δ3 intervals. Right, identity of new BW for cells that changed their BW over Δ3 intervals. Statistics: $\chi^2$, two-sided. **i** Mean initial whisker tuning curve for cells that maintained or changed tuning in a subsequent session. Data are pooled over Δ1, Δ2, and Δ3 (1270 pairs) and subsampled to ensure each cell was only represented once in each interval. Error bars are 95% CI over this subsampling. **j** Distribution of ΔCoM for neurons that had a significant tuning change. Statistics: KS, two-sided.

changed their tuning and responsiveness from session to session (example field: Fig. 5d, asterisks show tuning changes and arrows show unresponsive ↔ responsive transitions). We tracked two types of statistically significant tuning changes (Supplementary Fig. 7a, b). A change in the identity of the BW (ΔBW) was defined as appearance of a new strongest whisker that drove responses significantly greater than the prior BW, by permutation test. A change in tuning center-of-mass (ΔCoM) was defined as a shift in tuning CoM that was greater than a null distribution of ΔCoM obtained by shuffling trials between the two sessions (using bootstrapping with α = 0.05). Each cell was classified as having either stable tuning, ΔBW, or ΔCoM, with the latter indicating significant changes in tuning curve shape but no change in BW. This testing method identifies tuning changes that exceed measurement error on both sessions, accounting for both trial-to-trial variability and finite trial number.

We compared sensory tuning across Δ1, Δ2, and Δ3-session intervals (5.2 ± 1.2, 12.4 ± 1.3, and 18.7 ± 1.5 days). 25% of cells were whisker-responsive on any given day. Across Δ1 interval, 61% of neurons were unresponsive in both sessions, 25% wavered between responsive and non-responsive, and 14% were responsive in both sessions. The same was true for Δ2 and Δ3 (Fig. 5e, f). Of cells that were responsive in both sessions, many exhibited significant tuning changes (Fig. 5g). We analyzed tuning stability for neurons that were responsive in both sessions across Δ1, Δ2, and Δ3 intervals (n = 468, 419, and 389 cells). 35% of neurons exhibited a significant change in either BW or tuning CoM across Δ1 intervals, increasing to 46% of PYR cells for Δ3 intervals. In contrast, tuning changes assessed between alternate trials of the same session, or first and second halves of the same session, were near the expected false positive rate of 5% (Fig. 5h). When a neuron's BW changed, it typically shifted to an adjacent whisker in the same row or arc (Fig. 5h, inset). This same pattern was true for cells that were responsive in all 4 sessions (Supplemental Fig. 7c). Thus, nearly half of PYR cells underwent spontaneous tuning changes in expert mice over 2–3-weeks. Tuning changes did not reflect poor initial tuning, because unstably and stably tuned neurons both showed sharp tuning in the first session (Fig. 5i).

Together, variable tuning and variable responsiveness were substantial: Of 2771 neurons that were tracked across all 4 sessions, only 7% (199) remained whisker responsive on all 4 sessions, of which only 39% (78) maintained consistent tuning throughout (Supplementary Fig. 7d). We tested whether wavering responsiveness was related to tuning instability by examining wavering neurons that were initially responsive, became non-responsive, and then became responsive again over 3 sessions. 47% of these neurons (122/258 cells) significantly changed their BW or CoM, compared to 43% of neurons (307/710) that were stably responsive over these sessions (p = 0.273, Fisher's exact test). Thus, tuning instability is independent of wavering responsiveness.

To test whether tuning instability represents unbounded random drift, we compared the magnitude of ΔCoM between Δ1, Δ2, and Δ3 intervals. Random drift would cause ΔCoM magnitude to increase steadily with time[19]. However, ΔCoM magnitude for the Δ1

interval was equivalent to those for Δ2 and Δ3 intervals, when calculated either for unstably tuned neurons (Fig. 5j) or for all neurons (Supplementary Fig. 7e). Thus, tuning instability is prevalent and rapid, but is constrained.

## Tuning instability is structured within the whisker map
Tuning drift in non-topographic cortical representations has no clear relationship to circuit architecture and is often considered random[19]. In S1, a mechanism must exist to maintain average global map topography despite single-neuron tuning instability. This could occur if tuning instability is structured within the map. To test this, we analyzed tuning changes for neurons at different locations within the tuning ensemble, either CW-tuned neurons (in the columnar core of the ensemble) or non-CW tuned neurons (in the surround of the ensemble). Neurons that were tuned to non-CW whiskers in session 1 were substantially more prone to change tuning on subsequent sessions than CW-tuned neurons, and particularly to change BW (Fig. 6a). Most tuning change occurred for neurons that were located one column away from their BW column (Fig. 6a, right). This pattern was similar for Δ1, Δ2, or Δ3 intervals (Supplementary Fig. 8a, d), and is illustrated for all cells at Δ1 in one example mouse in Fig. 6b.

Factors besides map location predicted tuning stability only weakly, or not at all (Fig. 6c–e). BW response magnitude in session 1 did not predict tuning stability. Broadly tuned neurons were more likely to change tuning, while neurons located within local co-tuned clusters (defined by a 'cluster index' that measures the mean tuning similarity between a neuron and its neighbors within 20 μm) were slightly more likely to maintain tuning. Tuning instability was strongly related to map position, independent of whether neurons were located in columnar vs. septal compartments in L2/3 (Fig. 6f).

Thus, tuning instability is organized within the L2/3 map, with CW-tuned neurons having relatively stable tuning, while intermixed non-CW tuned neurons exhibit pronounced tuning instability. This rules out the possibility that tuning changes reflect day-to-day variation in behavioral state, or random measurement error. In contrast, changes in responsiveness were only weakly related to cell position in the whisker map, and instead were most strongly related to initial whisker-evoked response magnitude on the first session (Fig. 6g-i). To quantify these relationships, we trained a classifier to predict a cell's change in tuning or change in responsiveness from its initial response strength, tuning sharpness, cluster membership, and map location. The strongest factor in predicting tuning change and responsiveness change were map location and initial responsiveness, respectively (Supplementary Fig. 8f-g). Thus, tuning instability is localized to the non-columnar surround of each whisker's tuning ensemble (Fig. 6j).

## Disrupting whisker experience does not alter tuning instability
Tuning instability could reflect sensory-driven plasticity from experience outside the task, or internal variability driven by ongoing synaptic turnover or other internal processes[12,14]. Because L2/3 of sensory cortex exhibits robust plasticity in response to sensory input

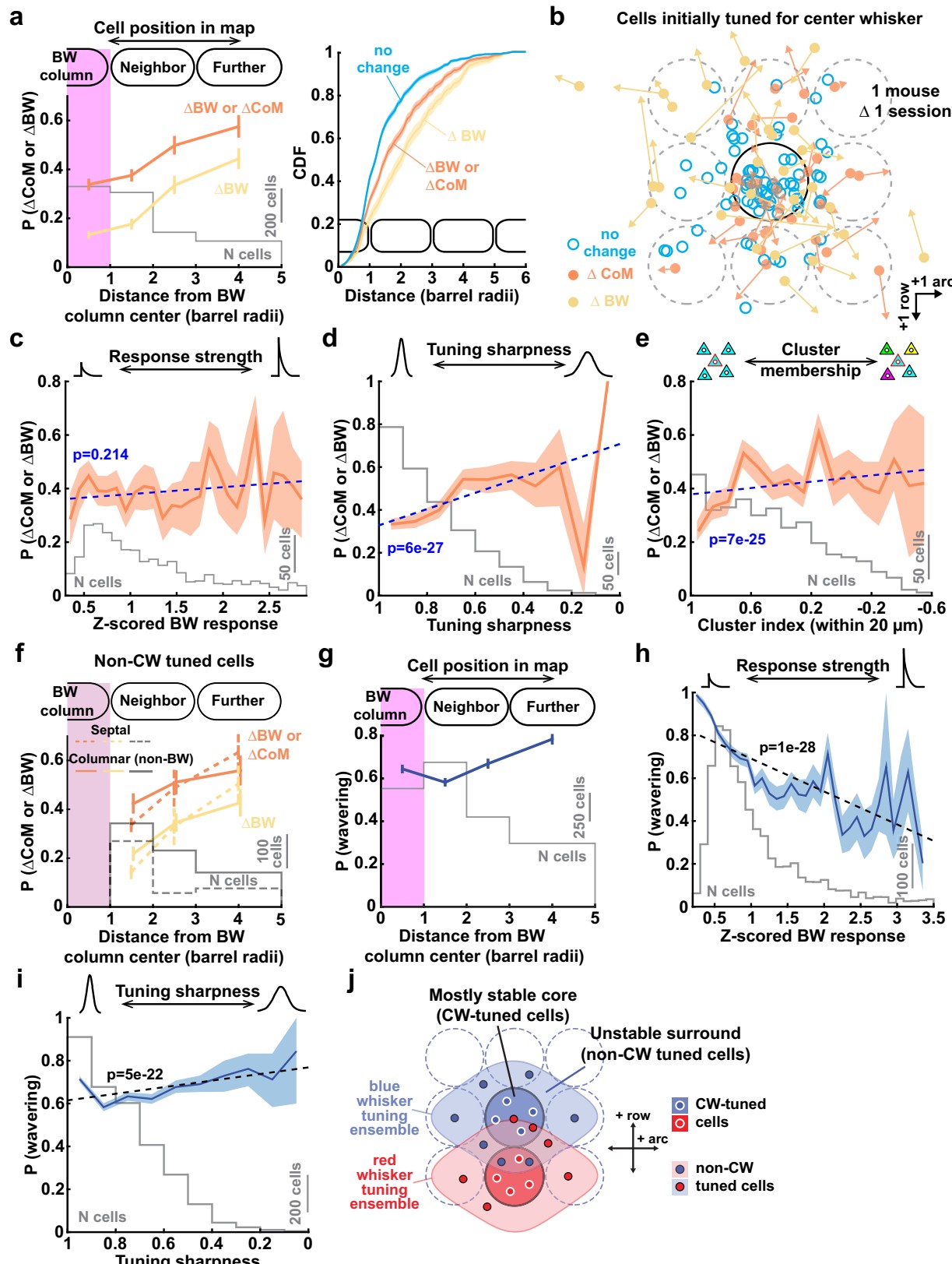

patterns[22,25,46,47], we tested whether instability was related to patterns of whisker sensation. After the 4 longitudinal sessions described above, we disrupted active whisking by injecting Botox into the mystacial pad. This substantially changes sensory statistics, and reduces but does not abolish whisker input. Whisking ceased within hours after injection, and paralysis was maintained over the next 3 weeks, during

which we performed 4 additional imaging sessions 4-7 days apart, tracking the same neurons as before Botox (Fig. 7a). Behavioral performance was not disrupted by Botox (Fig. 5b). During Botox sessions, the fraction of whisker-responsive neurons was similar to pre-Botox (Fig. 7b). Responsive neurons showed the same probability of tuning changes across Δ1, Δ2 or Δ3 Botox sessions as for pre-Botox (Fig. 7c).

**Fig. 6 | Tuning instability is structured within the L2/3 whisker map. a** Spatial organization of tuning instability. Left: mean fraction of neurons exhibiting a CoM change or a BW change, as a function of cell distance to its BW column center in session 1. Cells in the magenta area were initially CW-tuned. Right: Spatial distribution of cells (based on Session 1) in each tuning stability group. Data pooled across Δ1, Δ2, and Δ3 intervals (1270 pairs). Same dataset also in (**c–e**). **b** Tuning dynamics over Δ1 session (4–7 day interval), for all neurons from 4 imaging fields in one example mouse. Circles show cell location relative to the BW column in Session 1. Black circle: BW column boundary. Gray dash: Surrounding columns. Arrows show ΔCoM for each cell with a significant BW or CoM change. **c** Probability of tuning change as a function of initial BW response magnitude. Dashed line: linear regression. Statistics: two-sided t-statistic for slope ≠ zero. **d** Probability of tuning change as a function of initial tuning sharpness. Conventions as in (**c**). **e** Probability of tuning change as a function of initial cluster index in Session 1. Conventions as in (**c**). **f** Tuning stability for non-CW cells (in Session 1), separated into cells overlying septa (289 cells) vs. cells within any whisker column (179 cells). Conventions as in (**a**). **g** Fraction of neurons that altered responsiveness as a function of initial cell location relative to its BW column. Conventions as in (**a**). Data pooled across Δ1, Δ2, and Δ3 intervals (2430 pairs). Same dataset also in (**h,i**). **h** Probability of changes in responsiveness as a function of initial BW response magnitude. Dashed line: linear regression. Statistics: two-sided t-statistic for slope ≠ zero. **i** Probability of changes in responsiveness as a function of initial tuning sharpness. Conventions as in (**h**). **j** Conceptual model of the stable core and unstable surround compartments within each tuning ensemble in L2/3. All lines are means and shading or error bars are bootstrapped 95% CI after subsampling to ensure each cell is represented only once.

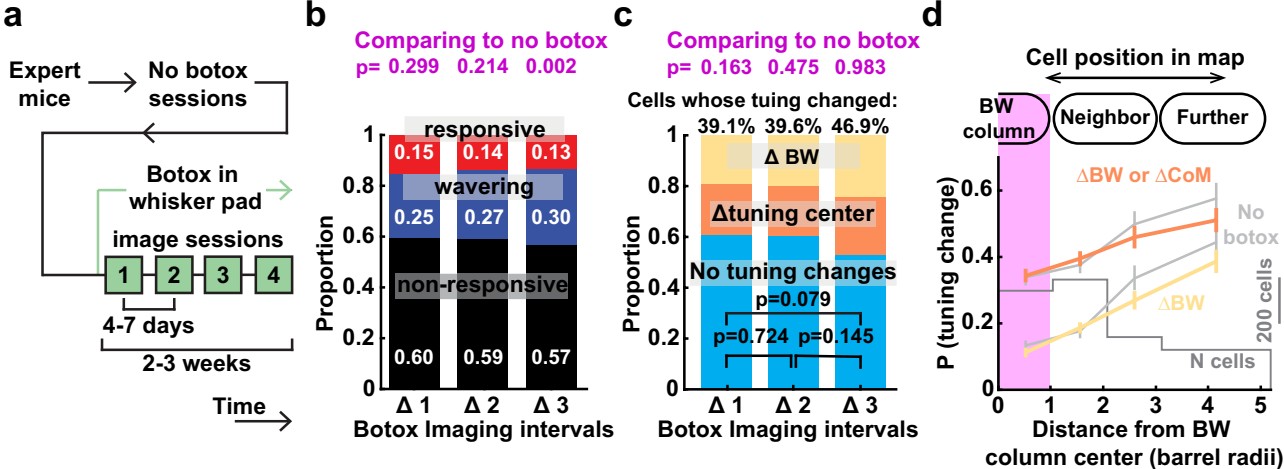

**Fig. 7 | Disrupting active whisking does not alter tuning instability. a** Timeline of longitudinal imaging from one imaging field after Botox injection to induce whisker paralysis. **b** Probability of changing whisker responsiveness for L2/3 PYR neurons tracked across Botox sessions. Top: *p*-values comparing to no-Botox data ($\chi^2$, two-sided). **c** Probability of changing whisker tuning for responsive neurons tracked across Botox sessions. Top: *p*-values comparing to no-Botox data ($\chi^2$, two-sided). **d** Spatial organization of tuning instability in Botox sessions (Δ1, Δ2, and Δ3 intervals pooled), analyzed as in Fig. 6a. Corresponding data from no-Botox sessions are shown in gray for comparison. Lines and error bars show mean and bootstrapped 95% CI after subsampling to ensure each cell is represented once in the dataset.

Nor did we detect any transient changes in instability at the onset of Botox, as might be expected if instability were driven by the onset of altered experience (Supplementary Fig. 8a, b). The topographic organization of tuning instability was also the same as pre-Botox (Fig. 7d and Supplementary Fig. 8c–e).

Thus, tuning instability does not require normal patterns of whisker experience outside the task. This also confirms that tuning instability is not due to whisker self-movement during the task. Because whisker paralysis disrupts sensory correlations and reduces whisker input, this suggests either that minimal, abnormal sensory input is sufficient to drive tuning instability, or that an internal source drives tuning instability (see Discussion).

### Individual whisker discrimination increases tuning stability

The one-versus-all task does not require discrimination between individual whiskers, and thus may not require highly distinct or stable single-whisker representations. Does this explain the single-whisker tuning instability that we observed? To test this, we trained Drd3-Cre:TIGRE2.0-GCaMP6s mice (*n* = 2) on a whisker identity discrimination task. On each trial, mice were presented with one of 9 single-whisker deflections (C1-3, D1-3, or E1-3) or a blank, with equal probability. Mice were trained to lick to D1, D2, or D3 whiskers (rewarded), but not to C1-3, E1-3, or blanks (unrewarded) (Fig. 8a). A 1-sec delay period separated stimulus presentation from the reward window, to prevent lick contamination of sensory responses. Mice took

21–23 days to become expert on this D-versus-C/E discrimination task after introducing all stimuli, and then showed stable performance over 10–14 days (d-prime = 1.37 and 1.44) (Fig. 8b). We performed longitudinal imaging in expert mice (3 imaging fields per mouse, each field imaged for 3 sessions, 4–7 days apart).

Training on this task caused several changes in mean tuning of L2/3 PYR cells in S1. In D columns, a higher fraction of cells was CW-tuned (and fewer cells were non-CW tuned) compared to the one-versus-all task (Fig. 8c). CW-tuned neurons within D columns showed stronger whisker responses and sharper tuning than the one-versus-all task (Fig. 8d, e). These effects were absent for C- or E-whisker tuned neurons in the same D columns (Supplemental Fig. 10). In C and E columns, there was a more modest effect on fraction of CW-tuned cells, and CW-tuned cells in these columns did not show stronger responses or sharper tuning (Supplemental Fig. 10). Thus, discrimination training had an effect on mean map structure, with L2/3 cells in D columns exhibiting more homogeneous columnar organization and stronger, sharper tuning for CWs than in one-versus-all expert mice.

D-vs-C/E discrimination reduced tuning instability for all cells in all columns, but particularly strongly for CW-tuned cells in D columns, relative to one-versus-all trained mice (Fig. 8f). To test whether CW- or non-CW tuned neurons were preferentially stabilized, we identified these cells by their tuning in the first session, and examined their tuning instability across Δ1 and Δ2 session intervals. Increased tuning stability was apparent exclusively for CW-tuned

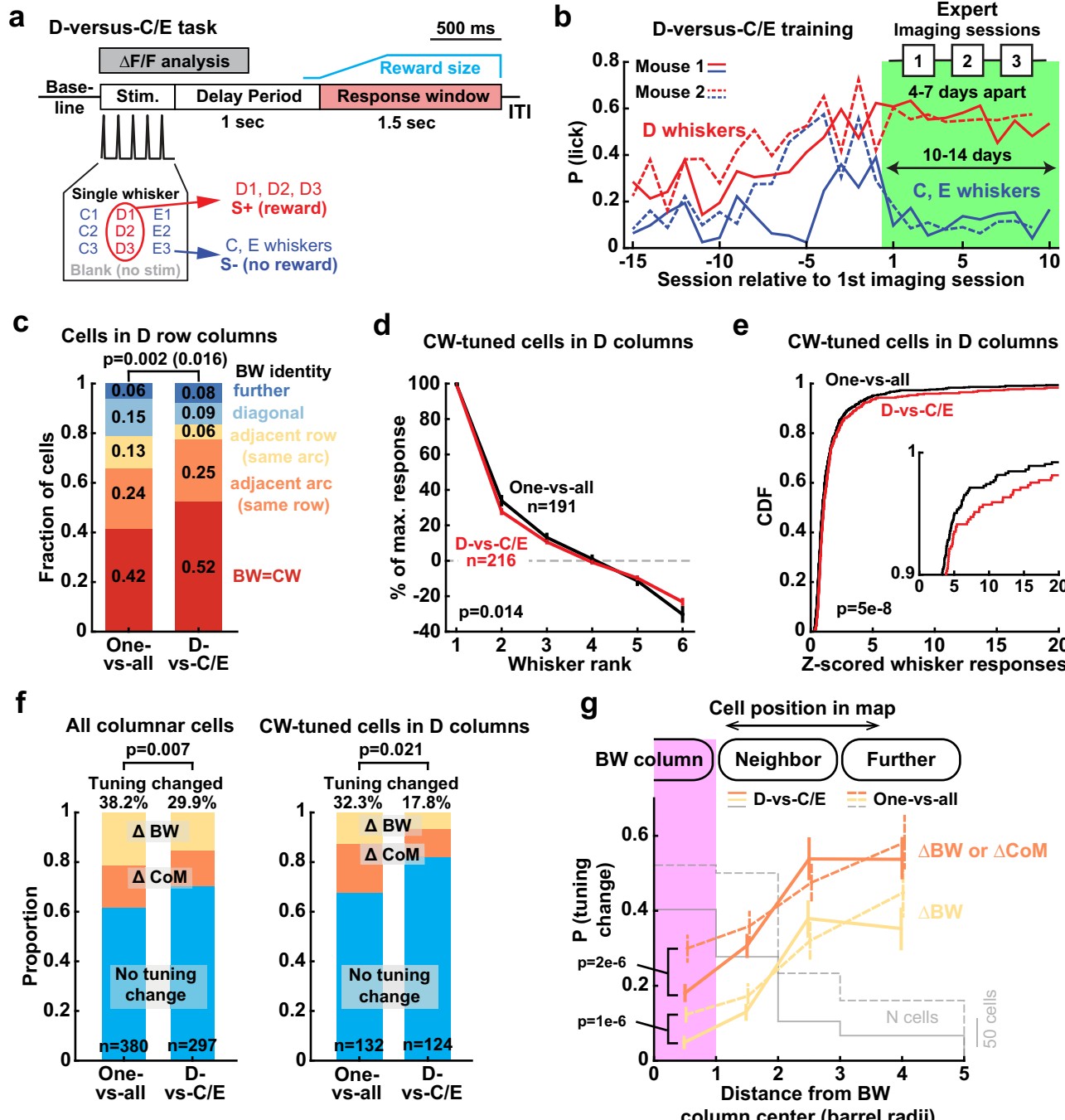

**Fig. 8 | D-versus-C/E discrimination training reduces tuning instability, but only for CW-tuned neurons. a** Trial structure and stimulus-reward contingencies for the D-versus-C/E whisker discrimination task. **b** Behavioral performance for 2 mice, during task learning (last 15 days before imaging) and during imaging sessions when mice were experts. **c** Tuning heterogeneity for cells in D-row columns, compared between one-versus-all mice ($n = 191$ cells from 4 mice) and D-versus-C/E mice ($n = 216$ cells from 2 mice). Statistics: $\chi^2$ for difference in distribution, and (in parentheses) Fisher's exact test for proportion of CW-tuned cells, two-sided. **d** Mean rank-ordered whisker tuning curves for CW-tuned neurons in D whisker columns. *p*-value is for task factor in unbalanced two-way ANOVA. Error bars: SEM.

**e** CW response magnitude for the same cells as in (**d**). Statistics: KS, two-sided. Inset shows top 10% responsive cells. **f** Proportion of responsive cells whose tuning changed significantly over imaging sessions, for all cells in all columns (left), and CW-tuned cells in D row columns (right). Data are pooled over Δ1 and Δ2 intervals. *n*: cells. Statistics: $\chi^2$, two-sided. **g** Spatial organization of tuning instability in D-versus-C/E discrimination experts (Δ1 and Δ2 intervals pooled), analyzed as in Fig. 6a. Corresponding data from the one-versus-all task are shown in dashed lines. Error bars show bootstrapped 95% confidence interval after subsampling to ensure each cell is represented once in the dataset. Statistics: rank-sum test between the two tasks within each bin, two-sided.

neurons, while tuning instability for non-CW tuned neurons was unchanged relative to one-versus-all mice (Fig. 8g). Thus, whisker discrimination training strengthened, sharpened, and stabilized tuning for CW-tuned neurons in D columns, but did not alter tuning instability among non-CW neurons.

## Discussion

Our findings show that in awake mice, L2/3 PYR neurons tuned for different whiskers are intermixed in each column, matching the salt-and-pepper organization in anesthetized animals[3,4,7,8,10] and as inferred from stimulating fewer whiskers in awake studies[2,5,22]. In L2/3, CW- and

non-CW tuned PYR neurons are intermixed in each column, and the set of neurons tuned for a given whisker is spatially dispersed, including both a columnar core of CW-tuned cells in that whisker's anatomical column, and a non-columnar surround of non-CW tuned cells that spills broadly into adjacent columns (Fig. 2). Our results show that these CW- and non-CW tuned cells are functionally distinct, differing in tuning stability, trial-to-trial reliability, tuning sharpness, and modulation by task demands (Figs. 1, 6, and 8). CW-tuned cells are also more likely to project to S2, and non-CW cells are more likely to exhibit multi-whisker combination tuning[3,8]. Thus, CW-tuned cells and non-CW tuned cells comprise functionally divergent core and surround compartments within the L2/3 whisker map. This differs from the view of L2/3 as simply an imprecise map with high local scatter[2,3,10].

Within each column, in addition to the known subcolumnar tuning gradient[2] we identified local clusters of co-tuned neurons (<20 μm apart). Tuning clusters have been observed for orientation tuning in V1 on the 5–40 μm scale[38,39], possibly reflecting microcolumns with shared synaptic connections and developmental origins[48], and for whisker direction tuning in S1[49]. Clustering does not occur for whisking vs. touch neurons[37] or for single-whisker vs. multi-whisker touch neurons, which instead segregate in column centers vs. edges[50].

Whether neural coding is stable or unstable in primary sensory cortex has been controversial. Higher order, non-topographic areas like hippocampus, posterior parietal cortex, or piriform cortex can show pronounced representational drift despite stable behavior[17–19]. But in topographic sensory areas like visual cortex[20,21,26,29,30] and S1[22,24,37], tuning for local sensory features has been considered to be largely stable in adults except during sensory learning[23,37,51]. This has been interpreted to suggest that cortical input stages are anatomically yoked to topographic peripheral input, forcing relative tuning stability, while higher order areas construct sparse, malleable population codes that store information in distributed ensembles that are ideal for flexible learning, but also drift[12,19]. Such drift poses challenges to stable perception and behavior, and likely requires downstream changes to stably read information from the drifting code[12]. Alternatively, drift may be confined to non-coding dimensions of population activity, so information readout is preserved[12,15–17].

Tuning instability was prominent in expert mice with stable behavior on the one-versus-all task, with ~46% of L2/3 PYR cells showing significant tuning changes and 23% completely changing BW over 5-18-days. This contrasts with prior reports of stable tuning for whisker direction, vibration frequency, and whisker response magnitude in S1[22,24,37], though one study suggested instability of 2-whisker preference[22]. The magnitude of tuning change (ΔCoM) did not increase from Δ1 to Δ3 sessions, and thus represents bounded instability, not unbounded tuning drift[19,44,52,53]. Several V1 studies, while they concluded that tuning was generally stable, in fact observed >20% of L2/3 neurons with session-to-session changes in ocular dominance, orientation, or visual object tuning that exceeded measurement error[16,23,25,29] and which also represent bounded instability[25]. Thus, coding instability already occurs in L2/3 of S1 and V1, but as bounded instability rather than unbounded drift.

Despite the general expectation that tuning instability occurs randomly within maps[19], we found that instability was preferentially localized to non-CW tuned neurons, while intermixed CW-tuned neurons had largely stable tuning. This was particularly evident in D-versus-C/E expert mice, where CW-tuned cells showed no BW changes above the expected 5% false-positive rate, while non-CW cells showed substantial BW tuning instability (Fig. 8g). This rules out artifactual sources of tuning instability in day-to-day behavioral variability or inconsistent sensory stimuli, which is more difficult to exclude in other systems[11,19]. Thus, the L2/3 map is composed of a stable, topographic, columnar core of cells for each whisker, surrounded by an unstable, poorly topographic surround whose tuning dances, apparently unpredictably, over days (Fig. 6j).

We tested two models of the origin of tuning instability. First, instability could reflect ongoing experience-dependent plasticity, driven by patterns of whisker experience outside the task[19]. If so, Botox whisker paralysis should either reduce instability (because paralysis reduces patterned whisker experience) or increase instability (because paralysis alters ongoing sensory statistics). But Botox produced no change in tuning instability prevalence or structure in the L2/3 map, either acutely or chronically over weeks. Thus, tuning instability is robust to external sensory patterns, suggesting that it may be internally generated, e.g. by ongoing synaptic turnover or memory consolidation[13,54]. This hypothesis will need to be tested more specifically in future studies.

Second, tuning instability could relate to sensory processing demands. The one-versus-all task does not require single-whisker discrimination, so somatotopic tuning instability may not impair behavior. When mice were trained to be experts on the D-versus-C/E task, which does require single whisker discrimination, CW-tuned cells in D columns became more prevalent and showed stronger tuning for D whiskers relative to one-vs-all expert mice, indicating a more precise columnar representation. Tuning instability was sharply reduced relative to one-versus-all mice, both broadly across all cells in all columns and even more strongly for CW-tuned cells in D columns. This increased stability was localized not to non-CW neurons, but to CW-tuned neurons, where it reduced instability substantially (Fig. 8g). As a result, in D-versus-C/E expert mice, tuning instability was even more localized to the non-CW compartment. Stabilized tuning for CW-tuned neurons predicts more stable population coding of whisker identity, beneficial for task performance. Thus, the prevalence of tuning instability in CW-tuned cells is related to behavioral demands. In contrast, the origin and function of tuning instability among non-CW cells remain unclear.

Overall, tuning instability in the non-CW compartment was a robust feature in one-versus-all expert mice, Botox-treated mice, and D-versus-C/E expert mice, suggesting it is an inherent property of the L2/3 map. We hypothesize that the relatively stable core compartment carries single-whisker information for touch localization, while the surround compartment carries more integrative or context-dependent information including for multi-whisker features[8,50,55], and drifts in its single-whisker tuning. Other salt-and-pepper maps, such as for tonotopy in the auditory cortex[56,57] and retinotopy in the visual cortex[58], may also exhibit distinct stable vs. more integrative, unstable compartments.

Tuning instability, even if widespread across neurons, will not drive perceptual instability if population activity changes are orthogonal to relevant population coding dimensions[16,17]. To implement this principle in S1, CW-tuned ensembles may be strongly weighted for downstream spatial discrimination, while non-CW neurons may be weighted for other aspects of whisker coding. Alternatively, unstable non-CW tuned cells may provide an unstable population code to downstream areas. Theory predicts that downstream areas can adapt to a drifting neural code if a stable internal model is also present that can guide training by Hebbian mechanisms[14]. Both CW-tuned and non-CW tuned neurons project downstream to M1 and S2[3]. Thus, CW-tuned neurons may provide a stable internal model to downstream regions to allow them to continue to stably interpret changing neural codes for whisker input. In this model, CW-tuned neurons could provide an anchor not only for the average whisker map in S1, but for downstream regions to deal with drifting codes at interior nodes of the cortical hierarchy.

## Methods
### Animals
Procedures were approved by the UC Berkeley Animal Care and Use Committee, and followed NIH guidelines. 3 mouse genotypes and GCaMP6s expression strategies were used. First, 11 Emx1-IRES-Cre mice

(JAX 005628) were used with viral expression of Cre-dependent GCaMP6s, which expressed GCaMP6s in all cortical excitatory cells. 6 of these mice were used for whisker-cued experiments and 5 for sound-cued experiments. Second, 6 mice transgenically expressing GCaMP6s in L2/3 PYR neurons were generated by crossing L2/3 PYR-specific Drd3-Cre mice (Tg(Drd3-cre)KI196Gsat[59], MMRRC_034610-UCD) with TIGRE2.0-GCaMP6s/Ai162 mice (TIT2L-GC6s-ICL-tTA2[32], JAX 031562). These mice had stable GCaMP6s expression which allowed them to be used for longitudinal imaging and Botox experiments. We did not use Emx1-IRES-Cre driven transgenic expression of GCaMP6s because this can lead to epilepsy[60]. Both viral GCaMP/Emx-1 Cre mice and transgenic GCaMP/Drd3-Cre mice showed salt-and-pepper organization with only modest differences (Supplementary Fig. 4). Because of these slight differences in map structure, all comparisons of whisker-cued and sound-cued effects used viral GCaMP/Emx-1 Cre mice, and all longitudinal imaging and Botox measurements used transgenic GCaMP/Drd3-Cre mice. To describe basic tuning properties (Figs. 1–3), we pooled both viral and transgenic GCaMP6s data. Finally, 2 C57BL/6J mice were used for AAV-driven pan-neuronal expression of nucleus-targeted GCaMP6s. Mice were of either sex. Drd3-Cre:TIGRE2.0 mice were on mixed background; Emx1-IRES-Cre mice and mice used for nucleus-targeted GCaMP6s expression were C57/B6J strain. Before surgery, mice were housed in cohorts of five or fewer, with running wheels (Mouse Igloo #K3327, Bio-Serv) in reverse 12/12 light-dark cycle with humidity 30-70% and temperature 20-26 °C, and all behavior training and experiments were conducted in the dark (active) cycle. Mice were singly housed in separate cages after craniotomy surgery.

## Surgery and viral injection

Mice (2.5–3 months old) were anesthetized with isoflurane (1–1.5% in $O_2$), and a stainless-steel head holder with 6 mm aperture was affixed to the skull using cyanoacrylate glue and dental cement. D1, D2 and D3 whisker columns were localized using transcranial intrinsic signal optical imaging[61,62]. A 3 mm diameter craniotomy was made centered on the D2 column. For viral GCaMP6s expression, AAV1-Syn-Flex-GCaMP6s-WPRE-SV40 (Addgene #100845-AAV1) was injected at 3–4 locations surrounding the D2 column at 250 μm and 350 μm depth. A chronic cranial window (3 mm diameter glass coverslip, #1 thickness, CS-3R, Warner Instrument) was attached with dental cement. Before surgery, mice received dexamethasone (2 mg/kg), enrofloxican (5 mg/kg), and meloxicam (10 mg/kg). Post-operative buprenorphine (0.1 mg/kg) was administered. For viral expression of nucleus-targeted GCaMP6s, mice were co-injected with AAV1-syn-H2B-GCaMP6s[43] (a gift from Dr. Na Ji, Depts. of Physics and Molecular & Cell Biology, University of California, Berkeley) and AAV1-mDlx-NLS-mRuby2 (Addgene #99130-AAV1), which drives mRuby2 expression to mark inhibitory interneurons[63].

## Behavioral tasks

At the start of each daily behavior session, mice were transiently anesthetized with isoflurane and head-fixed under the 2-photon microscope. 9 whiskers (rows C-E, arcs 1-3) were inserted into a 3 ×3 array of calibrated piezoelectric actuators, centered on the D2 whisker. Whiskers were not trimmed, and were threaded into tubes on the piezos, held by soft glue. Deflections were applied 5 mm from the face. A drink port with capacitive lick sensor recorded licks. Paw guards prevented paw contact with whiskers, piezos, or drink port. After whisker insertion, mice recovered from anesthesia and began the behavioral task. Mice had to suppress licking to initiate a behavioral trial. Training was performed in total visual darkness (using 850 nm IR illumination for behavioral monitoring). Uniform white noise (77.4 ± 0.5 dB) was continuously applied to mask sounds from piezo actuators and drink port opening (drink port: 58.9 ± 0.7 dB; all-whisker deflection: 69.2 ± 0.7 dB; single whisker deflection: 59.7 ± 0.5 dB;

background noise floor without white noise: 58.7 ± 0.5 dB; all sound levels were measured at the location of animal's ears). Tasks were controlled by an Arduino Mega 2560 and custom routines in Igor Pro (WaveMetrics).

**One-versus-all whisker discrimination task.** Each trial contained a 0.5 s baseline period, 0.5 s stimulus period, and 1.5 s response window. One randomly chosen stimulus was applied per trial, either 1 of 9 single whisker deflections, all-whisker deflection, one of 2 tones, or a blank (no stimulus). Whisker stimuli were ramp-return rostrocaudal deflections (300 μm, 5 ms rise/fall time, 10 ms duration). A train of 5 deflections (100 ms interval) was used to reliably evoke GCaMP signals. The all-whisker stimulus was simultaneous deflection across all 9 whiskers. Tone stimuli were a single tone pip (2 or 8 kHz, 200 ms duration, 82.4 ± 0.2 dB) delivered from a nearby speaker. Each trial was followed by a 3 ± 1 s interval before the mouse could initiate the next trial. Thus, consecutive stimuli were separated by > 5.5 ± 1 s.

On S+ trials (all-whisker deflection), water reward (2–4 μl) was automatically dispensed 300 ms into the response window. Licking was not required to dispense reward. Water was not dispensed on S− trials. Licking above a threshold rate during the response window was defined as a lick response, and scored as a hit on an S+ trial and a false alarm (FA) on an S− trial. FAs and misses were not punished. Mice learned that S+ stimuli predicted reward, evidenced by licking in S+ trials prior to reward delivery (Supplementary Fig. 1c).

**Sound-cued task.** The structure of each trial was the same as one-versus-all whisker discrimination task, except the S+ stimulus was one of the two tones.

**D-versus-C/E whisker discrimination task.** Each trial contained a 0.5 s baseline period, 0.5 s stimulus period, 1 s delay period, and 1.5 s response window. One randomly chosen single whisker stimulus (1 of the 9 whiskers) or a blank was presented per trial. The mouse was rewarded if the D1, D2, or D3 whisker was presented, and not rewarded for any other whisker or blank. Mice had to withhold licking during the post-stimulus delay period, and then lick during the response window. Water reward was only dispensed in response to licks during the response window. In order to train the response window, the reward volume was linearly increased as a function of first lick time in the period 0-500 ms into the response window. For licks after this time, a constant maximal reward was delivered (Fig. 8a). No additional cues were given to indicate the delay period. This delay-dependent reward increment encouraged animals to withhold their licking following stimulus delivery, and to receive a larger water reward if they licked later. No punishment was given if animals licked during the delay period, but that trial was aborted (i.e., no reward was delivered) and excluded from analysis.

## Training stages

**One-versus-all whisker discrimination task and sound-cued task.** 1–2 weeks after cranial window implantation, water was regulated (daily water intake of 0.7-1.0 mL was calibrated individually to achieve 85% of ad lib body weight, and weight and health were monitored daily). In Stage 1 training, mice were acclimated to head-fixation and the water port. In Stage 2, mice learned to lick for water rewards (2–4 μl) cued by a blue LED mounted on the lick port. In Stage 3, S+/S− training was begun using all-whisker deflection, one tone stimulus, and blank trials. For whisker-cued mice, the all-whisker stimulus was the S+. For sound-cued mice, the tone was the S+. The blue LED was still presented at water delivery. Over days, mice learned to lick to the S+ stimulus, evidenced by an advance in lick timing from after LED onset to before LED onset, after the S+ stimulus (Supplementary Fig. 1c), as well as by a reduction in FA licks. This training stage continued until FA

rate fell below 50%, and >50% of licks occurred prior to the blue LED cue. In Stage 4, the final full behavioral task was implemented by introducing the other 10 S− stimuli and removing the blue LED cue.

Imaging sessions began when mice reached stable Stage 4 performance with >900 trials per session and about 10–20% S+ trials. Whiskers remained intact throughout the experiment.

**D-versus-C/E whisker discrimination task.** Stage 1 and Stage 2 were the same as above. In Stage3, S+/S− training was begun using all 9 single whisker stimuli and blank, with D1-D3 as rewarded stimuli (S+) and C1–C3, E1–E3, and blank as unrewarded (S−). All stimuli and blanks were presented with equal probability. The blue LED was still presented at water delivery. Over days, mice learned to lick to S+ stimuli, evidenced by an advance in lick timing from after LED onset to before LED onset. When >50% of licks occurred prior to the blue LED cue for 2 days, the LED cue was removed, and the reward increment was implemented to begin to train the delay period. To start, the reward increment ramp onset was at the end of the stimulus period and the maximum reward (end of the ramp) was 1 s later. Both the start and the end of the reward ramp were gradually shifted until ramp onset began 1 s after the end of the stimulus period and ramp duration was 0.5 s. During this process, we moved ramp onset in 100 ms steps, and evaluated the behavioral performance. If the mouse reached $d' > 0.5$ and less than half S+ were aborted due to early licks, or if performance remained stable for 3 days, we increased the delay of ramp onset by another 100 ms and repeated the behavior evaluation.

Imaging sessions began when ramp onset occurred 1 s after the end of the stimulus period (i.e., a 1-s delay period), less than half S+ were aborted, and $d' > 1$. Whiskers remained intact throughout the experiment.

## Two photon imaging
2-photon imaging took place 4-6 weeks after viral injection. Imaging was performed with a Moveable Objective Microscope (Sutter) and Chameleon Ultra II Ti:Sapphire mode-locked laser (Coherent). GCaMP6s and mRuby2 were excited at 920 μm. Scanning utilized one resonant scanner (RESSCAN-MOM, Sutter) and one galvo scanner (Cambridge Technology). Emission was collected through a 16X immersion objective (0.8 NA, N16XLWD-PF, Nikon), bandpass-filtered with dichroic mirrors (green: HQ 575/50, red: HQ 610/75, Chroma), and GaAsP photomultiplier tubes (H10770PA-40, Hamamatsu). Laser power at the sample was 30-75 mW. Serial single plane images (512 × 512 pixels, 150–275 μm below dura) were acquired at 7.5 Hz (30 Hz acquisition, 4-frame average) using ScanImage5.6[64] (Vidrio). Two different field of view sizes (305 μm × 305 μm or 406 μm × 406 μm) were used.

Each daily session comprised 900–1000 trials. 2-5 imaging fields were sampled in each mouse. We obtained 24 imaging fields from whisker-cued Emx1-Cre mice, 16 from whisker-cued Drd3-Cre mice, 20 from sound-cued Emx1-Cre mice, 10 from nucleus-targeted GCaMP6s mice, and 6 from D-versus-C/E discrimination task mice. For the one-versus-all task, longitudinal imaging was performed on 4 Drd3-Cre:Ai162 mice, with each field reimaged 4 times at 4–7-day intervals before Botox injection into the whisker pad, and 4 more times after Botox during whisker paralysis. For the D-versus-C/E task, longitudinal imaging was performed on 2 Drd3-Cre:Ai162 mice, with each field reimaged 3 times at 4–7-day intervals.

## Histological localization of imaging fields and cells
Imaged cells were localized relative to L4 barrel boundaries using post-hoc histology. A 2-photon z-stack was collected spanning from the L2/3 imaging plane to the pial surface, at the end of each imaging session. After experiments were complete, the brain was removed and fixed in 4% paraformaldehyde, and the cortex was flattened and sectioned parallel to the cortical surface. Sections (50 μm thickness) were stained for cytochrome oxidase activity, which reveals L4 barrels and surface blood vessels (Supplementary Fig. 1d, right). Imaging fields were aligned to column boundaries using blood vessels as landmarks (Supplementary Fig. 1d).

## Calcium imaging analysis
Analysis used the CaImAn[42] algorithm and custom Matlab routines unless stated otherwise.

**Imaging processing and ROI selection.** Movies were corrected for slow X–Y motion using NoRMCorre[65]. Substantial Z-axis movement was not observed and not corrected. Neuronal regions-of-interests (ROIs) were defined using CaImAn with default settings. The CaImAn algorithm recognized 80% of visible cells, and remaining cells were manually annotated using CaImAn's manually_refine_components function based on the average image. $\Delta F/F$ traces were extracted by CaImAn, with $F_0$ defined as the 25th percentile of the fluorescence distribution for that ROI. Only ROIs near stimulated whisker columns were analyzed (defined as ≤ 1.25 barrel radii from the centroid of a stimulated whisker column). The total number of imaged neurons were: viral GCaMP6s one-vs-all: 3956; transgenic GCaMP6s one-vs-all: 3437; viral GCaMP6s sound-cued: 3084; nucleus-targeted GCaMP6s one-vs-all: 3364; D-versus-C/E task: 2289.

We did not find evidence of GCaMP overexpression in our longitudinal imaging experiments. To assay this, we calculated the nucleus/cytosol (N/C) ratio of GCaMP intensity for 1329 cells in the final imaging session of all 6 Drd3-Cre:TIGRE2.0 mice. These cells were selected based on three criteria: (1) cells did not contact or spatially overlap with other cells; (2) diameter was > 9 μm, which ensured the optical section included the nucleus; and (3) No visible processes extended from the soma, so that the cytosol signal reflected somatic cytosol. From the average image of each cell, we calculated cytosolic intensity as the average intensity of the outermost 2-pixel ring of the cell's ROI mask, and nuclear intensity as the minimal pixel intensity within the ROI mask. Prior references for expected N/C ratio values were performed in vitro[66], and may not be applicable to in vivo conditions. Therefore, we tested for non-physiological N/C ratio values by comparing the distribution of N/C ratios for responsive versus non-responsive cells, and for CW-tuned versus non-CW-tuned cells. We found no significant differences (responsive or not: $p = 0.269$, KS test, two-sided; CW- or non-CW-tuned: $p = 0.258$, KS test, two-sided). Thus, variance in N/C ratio was unrelated to whisker responsiveness and tuning. This suggests that cumulative GCaMP expression, which increases N/C ratio, did not bias our results. We also tested whether cells in the last longitudinal imaging session had abnormally slow GCaMP decay kinetics, which is another sign of overexpression. We identified 3478 cells in the last imaging session of all 6 Drd3-Cre:-TIGRE2.0 mice that exhibited identifiable isolated $Ca^{2+}$ transients. The mean decay halftime was 825.4 ± 180.2 ms, which is similar to previous studies[31,67], indicating that GCaMP6s kinetics are in the expected healthy range, even after long-term imaging.

**Whisker-evoked responses and receptive fields.** To avoid lick contamination, $\Delta F/F$ responses were only analyzed on non-lick trials in the one-versus-all and sound-cued tasks, or on trials with no licks during the stimulus and delay period in the D-versus-C/E task. Stimulus-evoked $\Delta F/F$ was defined as mean $\Delta F/F$ (0–1000 ms after stimulus onset) minus mean baseline $\Delta F/F$ (0–500 ms pre-stimulus). To identify significant whisker responses, we used a permutation test for difference in mean $\Delta F/F$ for each whisker relative to blank trials. In each iteration of the permutation test, single-trial $\Delta F/F$ data were randomly shuffled between whisker S− and blank trials, and the difference in mean response between these shuffled trial sets was calculated. This was repeated 10,000 times to generate a null distribution. A measured whisker response was considered significant if it exceeded the 95th

percentile of this null distribution. *p*-values were corrected for multiple comparisons across all S− stimuli with false discovery rate 0.05 (Benjamini–Hochberg procedure[68]). A cell was considered whisker-responsive if ≥1 whisker induced a significant positive Δ*F/F* response. A single trial was defined as responsive if stimulus-evoked Δ*F/F* exceeded the mean plus one standard deviation of blank trials.

Previous in vivo imaging studies[37] have shown that whisker stimulation can drive a negative Δ*F/F* response (reduction below baseline) in some cells, likely reflecting inhibition[69]. In our data set, 28.4% of whisker responsive PYR neurons (i.e., cells with a significant positive Δ*F/F* response to ≥1 whisker) had a significant negative Δ*F/F* to at least one non-BW whisker. Negative Δ*F/F* responses were small and slow compared to positive Δ*F/F* responses. Negative Δ*F/F* responses were only included when analyzing rank-ordered tuning curve shape and columnar whisker dominance index (CWDI, Fig. 1e), but were replaced with zero for calculation of tuning sharpness and signal correlation. 25.9% of PYR neurons exhibited at least one negative significant Δ*F/F* responses and no positive Δ*F/F* responses. These cells were considered non-whisker responsive and not analyzed further.

**Tuning of individual neurons.** The best whisker (BW) was defined as the whisker that evoked the largest mean Δ*F/F* response and was significantly greater than blanks. Equivalent best whiskers (eBWs) were defined as whiskers with significant responses whose amplitudes were statistically equivalent to the BW by permutation test. For a cell to be classified as non-CW-tuned, the non-CW response had to be statistically greater than the CW response (i.e., the CW was not an eBW for this cell). BW tuning sharpness was defined as $(R_{BW}-R_W)/(R_{BW} + R_W)$, where $R_{BW}$ = mean Δ*F/F* to BW, and $R_W$ = averaged mean Δ*F/F* for all other whiskers (whiskers that evoked a negative response were considered as zero). Columnar whisker (CW) preference (Fig. 2g) was calculated similarly as $(R_{CW}-R_W)/(R_{CW} + R_W)$, where $R_{CW}$ = Δ*F/F* to the CW. Rank-ordered tuning curves were calculated by ranking each stimulus from strongest to weakest within each cell (normalizing to the blank or the strongest response for that cell) and then averaging ranked tuning curves across cells. This quantifies tuning sharpness around each cell's BW, independent of somatotopic organization. For rank-ordered tuning curves, only cells whose BW was the center whisker or a center-edge whisker in the piezo array were included. This ensures that the BW plus 5 or 8 immediate adjacent whisker responses were sampled. To calculate tuning center-of-mass (CoM), the 9 whiskers were assigned to Cartesian coordinates in a 3 × 3 grid, and CoM was calculated from the response amplitudes, with negative responses zeroed.

The CW dominance index (Fig. 1e) was defined as (response to CW - response to strongest SW)/|response to CW + response to strongest SW|. A cell that responded exclusively to the CW would have CWDI = 1. A cell that responded equally to CW and the strongest SW would have CWDI = 0. CWDI > 1 or CWDI < 0 indicate cells whose strongest SW response was negative, or cells whose CW response was negative, respectively. Because cells were classified as CW-tuned unless a non-CW elicited a statistically greater response than the CW, a small fraction of CW-tuned cells have a negative CWDI value (i.e., for these cells, a non-CW elicited a slightly, but not significantly, larger response than the CW).

Tuning measured from brief whisker deflection trains, as used in this study, may differ from tuning to single deflections.

**Normalized anatomical reference frame for spatial analysis across imaging fields.** To project cells into a common columnar coordinate system, ROI coordinates were transformed into a polar reference frame. We first drew a vector from the centroid of a reference column to the ROI. The normalized distance from ROI to column center was calculated as (measured distance)/(distance from column center to column edge along this vector). This gives units of barrel column radii. To determine the angular position for each ROI, vectors were drawn

connecting the centroid of each surrounding column to the centroid of the reference column. These vectors defined equally spaced 45° angles in reference space, and ROI angle was determined relative to these vectors.

**Spatial clustering of similarly tuned neurons.** To test for tuning clusters, we examined three measures of tuning similarity—signal correlation, Euclidian distance between tuning center-of-mass (CoM), and probability of sharing the same BW. Signal correlation for a pair of neurons was calculated as the Pearson's correlation coefficient between the mean responses to each single whisker stimulus, with negative responses zeroed. These measures assess slightly different aspects of tuning: Signal correlation is influenced by all whiskers equally; tuning CoM is driven primarily by the several strongest whiskers within the receptive field; and BW identity assesses only the strongest single whisker. Similarity in sensory responsiveness between 2 neurons was calculated as |BW response$_{neuron1}$− BWresponse$_{neuron2}$| /|BW response$_{neuron1}$ + BWresponse$_{neuron2}$|.

We calculated each measure for all simultaneously imaged, co-columnar pairs of PYR neurons (14399 pairs), as a function of distance between cells in 10 μm spatial bins. Averaging within each bin revealed the mean tuning similarity as a function of distance between cells. To test for significant spatial clustering relative to a null model of completely random intermixing of tuning properties, we compared these results to a shuffled dataset in which the locations of co-columnar neurons were randomly shuffled within each column, repeated 10,000 times to generate a null distribution.

To test if tuning similarity of neuron pairs within 20 μm exceeded the general subcolumnar tuning gradient, we first calculated the linear regression of measured data from 30 to 200 μm (which appeared linear by inspection), and extrapolated the linear regression to first 20 μm to obtain the expected value. Significant differences between real data and linear regression were determined by permutation test, which subsampled real data in each bin for 10,000 times, compared subsampled mean with expected value from linear regression, and corrected for multiple comparison across all bins with false discovery rate of 0.05.

**Analysis of tuning stability by longitudinal imaging.** ROIs were identified independently for each imaging session by CaImAn. ROIs that corresponded to the same neuron across sessions were registered manually based on the average image for each session. 80.6% of imaged neurons could be traced in at least 2 out of 4 sessions and 2771 neurons could be traced over all sessions. Neurons that could not be traced tended to be close to the imaging field edge and were obscured by image registration, or exhibited very low activity and thus did not appear in average image.

To assess tuning stability across sessions, we first tested for a statistically significant change in BW, by testing whether a new whisker evoked a significantly stronger mean Δ*F/F* than the prior BW, assessed by permutation test (this is equivalent to asking whether the prior BW is still an eBW). If the BW was stable, we next tested for change in CoM using a bootstrap method (Supplemental Fig. 7a). A null distribution of ΔCoM was created by shuffling individual trials of the same whisker between the two sessions and computing ΔCoM over 10,000 iterations. This models the ΔCoM distribution expected from random sampling, if single-trial responses for each whisker came from the same distribution across the two sessions. Tuning ΔCoM was considered stable if the measured ΔCoM was not different from the null distribution, with α = 0.05. A neuron was considered to have stable tuning if neither BW identity nor ΔCoM changed significantly.

When neurons were present across all 4 sessions, each neuron could contribute 3 different Δ1 measurements (1st → 2nd, 2nd → 3rd, and 3rd → 4th), 2 different Δ2 measurements (1st → 3rd, 2nd → 4th) and one Δ3 measurement (1st → 4th). To avoid overcounting the same cell in Δ1 and Δ2 measurements, we randomly subsampled a

single Δ1 or Δ2 value for each cell, repeated this 1000 times, and reported mean and 95% confidence interval for these measurements (error bars or shadings in Figs. 5i, j, 6, 7d, 8h and Supplementary Fig. 7e, 8a–e, 9c–e,). To compare tuning stability between one-versus-all and D-versus-C/E tasks, we only examined Δ1 and Δ2 measurements, because longitudinal imaging in the D-versus-C/E task was only performed over 3 sessions.

The relative contribution of cell location, tuning sharpness, response magnitude, and cluster index in predicting tuning stability and stability of whisker responsiveness were evaluated using a generalized linear regression model (Supplementary Fig. 8f-g). Binary outcome (did tuning change or remain stable? did whisker responsiveness change or remain stable?) was predicted by the 4 factors of cell location, tuning sharpness, response magnitude, and cluster index (for tuning change) or the first 3 factors only (for responsiveness change) using Matlab function glmfit(), using binomial distribution and logit link function, with 10-fold cross-validation.

**Validation that active whisker movement was minimal during behavior.** Whiskers contralateral to the piezo array were imaged at 15 Hz under 850 nm IR illumination during 2-p imaging sessions. Movement of two C-row whiskers (either C1–C2 or C2–C3) was measured in each mouse. Whisker position was measured along a curved trajectory ~2 cm from whisker pad, and instantaneous velocity was calculated from position. K-means clustering of velocity values was used to identify a threshold that separated active movement from non-movement within each movie. Using this threshold, we then calculated the percentage of non-lick whisker S− trials (i.e., trials that were used for 2p imaging analysis) that contained whisker movement. This value was averaged across the 2 whiskers to give a single value per mouse ($n$ = 5 mice).

**Spatial subsampling of ROIs for comparing whisker-cued (one-vs-all) and sound-cued imaging results.** The comparison of whisker-cued and sound-cued results in Fig. 4 includes all ROIs that were whisker responsive and located in or near the stimulated whisker columns, as defined above. We validated the functional differences between whisker-cued and sound-cued mice by performing additional analysis to correct for modest differences in spatial distribution of imaged neurons in whisker-cued and sound-cued datasets. To do so, we subsampled the data to generate spatially identical sampling in whisker-cued and sound-cued populations. In each iteration of subsampling, cells were randomly chosen from whisker-cued or sound-cued mice so that the numbers of cells within each whisker column were the same in whisker-cued and sound-cued mice. Data analysis was done for these subsampled cells. We performed 1000 iterations of this subsampling. The resulting mean and 95% confidence intervals are reported in Supplementary Fig. 6.

**Botox injection**
Botox (Botulinum Neurotoxin Type A, List Labs #130A) was reconstituted to a stock solution (20 ng/μl) with distilled water containing 1 mg/ml bovine serum albumin. To make injection solution, the stock was diluted with phosphate buffered saline to 10 pg/μl. Mystacial pads were subcutaneously injected with 1 μl of solution via microliter syringe (Hamilton). At first injection, mice stopped whisking within hours, and paralysis from a single injection lasted 7 days with full whisking recovering within 2 weeks. To achieve continuous paralysis across the Botox imaging period, we injected a supplemental 50% dose once per week. We inspected mice daily to confirm the absence of whisking.

**Statistics**
Statistical methods are described in Figure Legends and above. The sample size was not pre-determined. All tests were two-tailed except for permutation tests. Whisker-cued and sound-cued groups were not randomized, but were two sequential cohorts of mice. Single neurons were the unit N, except as follows: Mouse behavior was quantified by mouse and by behavioral session (Figs. 1b, 1c, 4b, 4c, and 5b, Supplementary Fig. 1b, c, g, h, 3b-c, and 8e). The fraction of responsive neurons per imaging field and spontaneous activity in longitudinal imaging were analyzed by imaging field (Fig. 4d, and 5c, Supplementary Fig. 4a).

In violin plots, circle is median, horizontal line is the mean, thick vertical line is interquartile range, and thin vertical line is 1.5x interquartile range (Figs. 1g and 4g, Supplementary Fig. 2c and 4c).

### Reporting summary
Further information on research design is available in the Nature Research Reporting Summary linked to this article.

## Data availability
Source data for the figures are provided with this paper. Processed imaging data for this study are available on the Feldman lab GitHub repository, https://github.com/dfeldman189/Wang2022Data. Source data are provided with this paper.

## Code availability
Matlab analysis code used for imaging data analysis are available on the Feldman lab GitHub repository, https://github.com/dfeldman189/Wang2022Data.

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

## Acknowledgements

This work was supported by R37 NS092367 from NIH. We thank Na Ji (UC Berkeley) for the nucleus-targeted GCaMP6s virus, and for discussions.

## Author contributions

H.C.W.and D.E.F. designed the study. H.C.W. performed the experiments and analyzed the data. A.M.L. provided analysis software. H.C.W. and D.E.F. wrote the manuscript.

## Competing interests

The authors declare no competing interests.
