## [Peer Review File · Nature Communications]

Tuning instability of non-columnar neurons in the salt-and-pepper whisker map in somatosensory cortexREVIEWER COMMENTS

Reviewer #1 (Remarks to the Author):

Wang et al investigate the whisker-specific tuning of neurons in superficial layers of primary somatosensory cortex in mice trained in a stimulus detection task. Because whisker-preference maps in L2/3 show a salt-and-pepper organization of whisker preference but L4 is dominated by its cognate whisker, the authors propose that this organization must be actively constructed by L2/3 neurons. Results largely match what has been described in anaesthetized animals, and some of this work (particularly that which was carried out using single-unit recordings) is not well-cited. Although the analysis is highly quantitative and generally rigorous, the major insights provided by the study are modest and better suited for a more specialized audience.

1. The main advance appears to be the analysis of maps in awake animals performing a task. However, the results appear largely similar to findings from anaesthetized mice. It has been well-established that L2/3 neurons show variable tuning, and that altered sensory experience can shift tuning preferences. The novelty of the work is reduced by the large body of prior studies, both cited and uncited, that show a similar mapping. Although the analysis of tuning stability, using the power of repeat imaging, is more novel, these results are mainly descriptive and their significance or how they revise current models is not well-developed. Is there any difference in the maps from awake animals compared to anaesthetized animals?
2. It is unclear what the authors mean by “active construction” of untuned neurons; passive receipt of sensory information and Hebbian mechanisms, given the interconnectivity of L2/3 neurons across barrel columns, might be sufficient to generate this effect. In addition, prior studies have established that nearby neurons are more likely to share connectivity; the significance of the 20 um distance for similarly-tuned neurons does not seem surprising. Thus, the conclusion that this instability is highly-structured seems overstated. In addition, the idea that some untuned cells serve to “actively explore novel sensory codes” seems to be unsupported by the data. What is the evidence that they are exploring a code? What code are they exploring – multiwhisker activation?
3. The abstract talks about expert mice, but there is no mention of training, or what the task actually is. The authors should make it clear that neural activity maps were characterized after extensive training in a whisker detection task in both the abstract and introduction. In Figure 1, is day 1 the first day of training? This is not clear from the figure legend, and indeed it is surprising if the mice performed at 80% from the first training day. Secondly, the mice appear to improve their accuracy over the imaging period as shown in Figure 1b. If this is the case, then it is less surprising that surround-tuned neurons adjust their preferred whisker across the imaging period, as behavior is still changing. Does the structure of the task (reward of multiwhisker stimulation) influence the characterized responses – perhaps if just two whiskers were stimulated, the representation of the second stimulated whisker would be stabilized? Could the use of a multiwhisker stimulus coupled to a reward drive the dynamic tuning of the neurons observed?

Minor

1. Emx1 mice show abnormal activity in Ca imaging experiments as well as altered dendritic structure and synapse organization. Were there differences in results from Emx and DRd3 animals? Also, what is the background of the Drd3 mice?
2. Was there any direction selectivity to the tuned and less-tuned neurons? Note that L2/3 in adult mice is exactly where directional maps (that show some local clustering) have been described. How deep was the imaging? Was there a difference between deeper and more superficial layers? One might expect that deep L3 would be better tuned than superficial L2.
3. What is the rationale for grouping all cells together, and not averaging across animals? What efforts were taken to ensure that there weren't more cells from one animal dominating the average? Do all animals show the same thing?
4. Wouldn't it be more accurate to say that tuning was more stable in CW, rather that instability was concentrated in non-CW neurons?
5. “ ... while non-CW neurons may be preferentially read out for coding of non-spatial features.” What are the non-spatial features that might be encoded? Is this velocity or frequency? Can the authors be

more precise about new experiments and predictions? This would help underscore the significance of the major findings, which otherwise seem very consistent with prior work.

Reviewer #2 (Remarks to the Author):

This paper examines long term single cell responses in somatosensory cortex (whisker barrel cortex). The authors find that a large proportion of primary neurons within S1 continually change their preferred stimulus. Interfering with active whisking does not substantially change this finding. The main findings are important because they challenge a number of long-standing hypotheses about the rigidity of cellular responses in primary sensory cortex. The work also complements a growing number of papers that have observed similar ongoing reconfiguration in the rest of the mammalian brain. It is an excellent study: the experiments are rigorous and the results are presented clearly for the most part and analysed appropriately. My only comments are on exposition. In general the paper is rather terse; the authors should use a few more words to explain what is going on and what it might mean. This, in addition to a fuller discussion of related literature (including theory) could improve it further by making the paper more accessible and connecting the findings to a wider and deeper literature on this topic.

Comments:

1. The role of auditory cue should be better motivated from the start. Why is it being used in a tactile task? Is the multisensory part particularly relevant to the investigation? It doesn't seem to be, based on the main findings and conclusions.
2. In describing the extraction of calcium signals, the phrase 'neuropil contamination' is a little misleading. Really you are saying that collateral projections may carry signals as well and that you are trying to restrict the analysis to the somata that are within the column. This point could be expanded because it raises questions about whether defining 'barrels' according to the location of somata really makes sense in the first place.
3. S1 is described as having prominent experience-dependent plasticity - maybe add a sentence to summarize the relevant type of plasticity that is observed?
4. The S- condition is not defined explicitly – help the readers more here.
5. The sentence introducing noise correlations - maybe parenthesize the phrase "noise correlations" and not "trial-by-trial covariability"? Many of us feel it is a lingering term that should be retired eventually and the alternative term the authors use is far superior!
6. Some of the writing in the main part of the results is very terse and might benefit from reordering, e.g. in the text surrounding Fig 5 I started off asking whether chronic imaging was being done in expert mice, only to find it later on in a very busy paragraph. This applies in the previous section too: it might be better to state upfront the goal of the upcoming set of results and whether each set of results is in trained mice, or mice that are learning to discriminate stimuli.
7. Fig 5b – it is hard to do the visual longitudinal comparison across these images. Would it be possible to overlay consecutive images somehow, or indicate changes with a marker/silhouette?
8. 5f - very busy and hard to parse. Bold red creates a weird optical illusion with the overlaid black text. This is one of the main summaries of one of the main findings, so it would pay to spend some time with this figure to see if it can be made more user friendly and easier on the eye. Ditto 7c.
9. In 1e, 5c - check alignment and sizing of subpanels in the figures. They are crowded and hard to look at in places.

10. The conclusion of botox experiments is a little strong. They can't rule out experience-driven plasticity, because whiskers can still transduce stimuli passively. And the botox injection and any subsequent adaptation could itself cause reorganisation of S1 responses over extended periods of time. There is no way to rule out this confound in the present study, so the conclusion should be appropriately mitigated.

11. The authors have cited a few important papers on representational drift, but a number of key references are missing. This is not just about having a 'complete list', there are several that posit the role of drift and the possible causes and these are important to contextualise the findings and show their connection with wider theories of how neural circuits maintain overall function. Lütcke, Margolis & Helmchen TINS (2013) is one of the earliest and most prescient reviews on the phenomenon being studied here. Rule, O'Leary & Harvey, Curr Opin in Neurobiology (2019) offers several specific hypotheses for the roles for drift that speak to the results in this manuscript. Similarly, Raman & O'Leary, eLife (2021) shows why plasticity processes that reorganise neural responses (at the synaptic level) are expected to persist in the absence of a learning signal - a prediction germane to the results around Figure 7. Finally there are a few theory papers that do a nice job of explaining how drift might coexist with stable circuit function: Druckmann & Chklovskii, Curr Biol (2012) somewhat anticipate this (they deal with fluctuations on fast timescales, but the high level point is there). Fauth, Wörgötter & Tetzlaff, PLoS Comp Biol (2015) present a model that maintains function despite synaptic turnover. Rule & O'Leary, bioRxiv (2021) show how a continually reconfiguring population representation can reliably interact with more stable representations in other circuits. This speaks to the idea that whisker barrels contain an anatomically rigid 'input' (e.g. in L4) while allowing a drifting representation in S1 as a whole.

Reviewer #3 (Remarks to the Author):

Comments on Wang et al. 2022

In this study Wang and his colleagues used 2P imaging to characterize the tuning properties of layer 2/3 neurons in the barrel cortex of awake mice, and asked how these tuning properties change over time, depend on attention, and are by whisking. They found a salt-and-paper organization of the cells, such that about 50% of cells that belong to a given barrel anatomically respond to other whiskers rather than the one that represents this barrel. They then asked if this organization depends on whether mice were attending to a whisker discrimination or to sound discrimination task and found no effect except for a small change in noise correlations among cells. The more interesting and novel findings were their discovery of anchor cells that kept their selectivity over several sessions (2-3 weeks) while other cells demonstrated a drift in their preferred whisker. Importantly, they found that stability was strongly related to the anatomical distance from the center of the barrel - cells that resided further away from the center of the barrel demonstrated greater drift. Finally, preventing whisking using Botox had a negligible effect on both the tuning properties and representational drift. The authors also used an additional type of GCaMP to show that neuropil signals were not the cause of local clustering among cells.

These findings are novel, important and illustrate representation and representational drifts in primary sensory areas in awake mice. In particular, the relationship between anatomy and drift (i.e., the dependency of the drift as function of the distance to the center of the barrel). Moreover, the experimental design and the methods are reproducible. Hence, with moderate revision this study should be accepted as it is clearly suitable to a large audience of system neuroscientists.

I would like to fully disclose that I reviewed an early version of this study for another journal and found that the authors greatly augmented the findings and increased the scope of the paper in the direction of plasticity. They also addressed several technical issues related to the imaging methods.

Moderate:

1. As I wrote in my earlier assessment of the paper, the authors have used repetitive stimulation of each whisker to evoke sensory response. Train stimulation leads to substantial adaptation and/or facilitation which clearly alters the receptive field profile of cells (adaptation is also known in awake mice, see Musall et al. 2014). This effect may alter the RF profile. For example, a given cell may adapt more to its corresponding whisker and less to an adjacent whisker. In this case the cell may exhibit stronger calcium signal to the adjacent whisker, despite responding less to the first stimulus in the train. Although I don't ask for additional experiments or analysis, this issue should be discussed.
2. The authors wrote that they did not observe over-expression, but there are a few clearly over-expressing cells in 1D (filled nuclei). Please exclude them from all the analyses, and clearly specify an exclusion criterion, e.g. using the N/C (nuclear/cytosolic) ratio as in Yang et al. 2018 (NC) or rise/decay dynamics.
3. It is important to see some additional analysis of the 'wavering' characteristics of some of the cells (i.e., cells that lost their whisker responsiveness completely or gained it from none) with respect to the cells' locations and their CW status, similar to the analysis that is done with the changes in BW or CoM. This is crucial as the fraction of such cells is even larger than the fraction of cells that were responsive across both sessions. Are wavering cells more likely to be septal or non-CW tuned than CW-tuned? What is the function between the wavering likelihood and the barrel center?
4. The authors wrote "GCaMP6s expression was stable throughout", but this should be supported with a figure or at least a calculation. To show GCaMP expression is stable throughout, plot the magnitude of baseline fluctuations (ongoing activity) for each cell over time, and check if there is an overall trend.
5. Can the authors check stability of the representations in the data that comes from the auditory task? It is well possible that drifts are different in animals that trained for the whisker task than naïve mice. Although doing additional experiments can be useful, I believe that looking at the data from the auditory task will be good enough to address this issue.

Minor:

1. In several places there is use of the word 'identical', e.g. "Identical local clustering of tuning similarity was observed" and "PYR cell response amplitude and tuning sharpness were identical in SC and WC mice". Please remove these uses and in each place state that you did not find a statistically significant difference.
2. You write "unstably and stably tuned neurons showed similar responsiveness and tuning sharpness in the first session" but the proportions look somewhat different. Please quantify as you do with KS or chi-squared for other comparisons.
3. More details on the masking noise are needed in the methods section.

Reviewer #4 (Remarks to the Author):

In this manuscript Wang and Colleagues carry out an elegant and well performed set of experiments to define the organisation of whisker tuning in sensory cortex, and how the spatial organisation of individual neurons with specific whisker tuning relates to the overall topographical organisation of sensory columns. The authors then go on to investigate the stability of this tuning and show that the spatial organisation of neurons with respect to columns correlates with stability of whisker tuning – where neurons tuned to whiskers outside their columnar location are less stable. The authors go on to propose that this property might be a crucial determinant for the balance between stability and flexibility during experience.

Overall this is a well performed set of experiments, and I enjoyed reading it – especially the rigour of the experiments and their analysis. My main concern is that although a nice set of experiments done well, it seems very descriptive. My worry is the manuscript does not fully convey the crucial biological implications of these findings, and the consequences of this organisation for circuit function is what would be more appropriate, rather than just the organisation itself. Some specific thoughts associated

with this are expanded on below, and could potentially be addressed with a substantial extra discussion, but ideally there would be at least some experimental evidence for what these findings might mean more broadly for the functioning of the circuit.

In addition, there are a few minor technical points that would be beneficial to be more openly discussed and presented.

Main points:

1. The authors propose that due to their more unstable dynamics, non CW tuned cells may play a preferential role in plasticity, compared to the more stable CW tuned cells acting as a 'backbone', but this is not tested (for example by monitoring the activity across learning, or across a contingency change). It feels like all of the experiments in the manuscript are leading up to that point, but then the experiment to explicitly test this is missing. In my opinion, the manuscript would be substantially strengthened if experimental or computational insight into the differential contribution of these two classes of cells could be provided.

2. The explanation of the relationship between the experiments and the task being performed was a bit vague, and could be laid out more clearly. To someone outside the project it is not immediately clear to what extent the findings are dependent on the specifics of the task – why is the task designed as it is, and how does this specifically help the analysis? More generally, as the authors are explicitly comparing their findings to anaesthetised recordings looking at passive whisker tuning, it would be good to explain the decision to investigate this in context of a learnt task? Is it simply to keep the mouse 'whisker attentive'? Could the specifics of the learnt task be altering the organisation and function of the circuit? This is addressed for the distribution finding with the sound task in Fig 4, but not for the stability of tuning, which is the main take home and potentially most related to behavioural strategy.

3. Related to above, from a learning point of view, the design of the task seems initially a bit lop sided, as in essence the mouse is being asked to identify 'all whiskers' vs 'one whisker' which could be interpreted as a threshold or summation task rather than a discrimination. It would be very beneficial to explicitly explain why this design was chosen – presumably to allow receptive field mapping of each whisker during CS- so as to control for reward artifacts? However in addition, it would be beneficial to have critical discussion of the alternative possibilities or specificity of the findings to the task. For example, could the organisation found be due to the specific demands of the task? In this specific task the discrimination depends on the ability to integrate stim from all whiskers (CS+) and compare to input from just one whisker (CS-) Could this contribute to the instability in 'outside column' neurons? As in the task the identity of the individual whisker is irrelevant to the learnt task, all that is needed to be known is if it is stimulated with others.

Technical points:

I think it is important to have some more explanation of why *drd3* and *emx* cre were used, and the potential caveats that might arise from imaging subpopulations of neurons when considering the distribution of cells in the circuit. This discussion could include a comparison to e.g. viral nuclear *gcamp* utilised in figure 3. Overall I think the fact that similar findings are shown across multiple expression strategies is a good thing – but it is worth discussing fully and openly.

Minor:

A personal preference, but there are a huge number of acronyms – I had to keep looking back to work out what was being talked about – I would recommend just spelling it out each time.

It would be very beneficial – especially as this is a new task – to present the data for learning of the task as well as just the days before expts started – how long did it take to get to criterion, and did learning rate correlate with measures studied?

Response to Reviewers

We thank the reviewers for their comments. To address them, we have added new experiments and new analysis, and substantially rewrote the text and figures.

We believe that our study provides an important advance in understanding the organization of the whisker sensory code in S1. By mapping receptive fields in awake mice more precisely than ever before, and by tracking tuning longitudinally over days and weeks in expert mice, we discovered that L2/3 is not simply a poorly topographic map with spatially intermixed whisker-related ensembles. Instead, L2/3 contains two functionally distinct compartments – a ‘columnar core’ of PYR neurons with stable tuning for the columnar whisker, and a ‘non-columnar surround’ of PYR neurons that are spatially intermixed, but tuned for non-columnar whiskers and whose tuning is markedly unstable over days. This tuning instability is the major novel finding, and provides a new view of L2/3 map organization.

We show that tuning instability is highly prevalent in task-expert mice with stable behavioral performance, and is largely confined to non-columnar surround neurons. We explore the origins of tuning instability, and show that, surprisingly, it is unchanged by major disruptions in sensory experience, suggesting that it may reflect an internally generated plasticity process rather than ongoing sensory-driven plasticity. In a new experiment (Figure 8), we test whether instability is related to behavioral use of whisker input, by comparing instability in mice trained to discriminate individual whiskers, vs. mice performing a one-vs-all task that does not require individual whisker discrimination. Mice discriminating individual whiskers show stabilized whisker tuning for many cells, although non-columnar surround neurons remained markedly unstable. Thus, instability is modulated by training on specific sensory tasks. Finally, we provide deeper discussion of theories of tuning instability and representational drift, and propose that columnar core neurons provide a stable internal somatotopic representation that allows downstream areas to adapt to the drifting neural code provided by unstable surround neurons, as proposed in a recent theoretical model of tuning drift ¹.

Reviewer #1

1. The main advance appears to be the analysis of maps in awake animals performing a task. However, the results appear largely similar to findings from anaesthetized mice. It has been well-established that L2/3 neurons show variable tuning, and that altered sensory experience can shift tuning preferences. The novelty of the work is reduced by the large body of prior studies, both cited and uncited, that show a similar mapping. Although the analysis of tuning stability, using the power of repeat imaging, is more novel, these results are mainly descriptive and their significance or how they revise current models is not well-developed. Is there any difference in the maps from awake animals compared to anaesthetized animals?

This is the first precise characterization of whisker receptive fields and map structure in awake, whisker-attentive mice, and we report two types of findings. First, basic map topography is salt-and-pepper intermixed, similar to anesthetized animals²⁻⁴ and a prior study in awake whisking mice⁵ (Figs. 1-2). We report a novel scale of somatotopic organization in the form of local clusters of co-tuned neurons within the salt-and-pepper map (Fig. 3). While somatotopic tuning clusters have not been observed previously, they may be related to clusters of neurons with similar direction tuning⁶. Thus, basic map topography is similar to what has been observed under anesthesia. But these are not the major findings of the paper.

The major findings are the new features of map organization that we discovered through longitudinal imaging. These include the high prevalence of tuning instability (Fig. 5), the fact that tuning instability is structured within the map (Fig. 6), and the fact that instability in the non-CW surround is highly robust to sensory experience and learning manipulations (Figs. 7-8). Together, these findings demonstrate that there are novel, functionally distinct compartments within the L2/3 map, shown schematically in Fig. 6j. This transforms our understanding of the L2/3 map from simply a

topographically imprecise map of intermixed tuning ensembles (similar to orientation or retinotopy in mouse V1) to a map in which each whisker is represented by a stable core of neurons with strong columnar topography, surrounded by a broad, non-columnar surround of neurons whose tuning dances unpredictably over days. We propose distinct functions for these novel columnar and non-columnar compartments. This is a major revision in understanding S1 organization (see Discussion).

The high prevalence of tuning instability in S1 challenges the standard notion that neural coding of local sensory features is stable in primary sensory cortex in the absence of new learning. Instead, the brain must have mechanisms for adapting to unstable codes, or for selectively ignoring unstable neurons (see Discussion). The fact that L2/3 sensory tuning is known to shift in response to experience does not predict the tuning instability we observe – in fact, we show that neither perturbation of sensory experience by whisker paralysis nor whisker-reward pairing in a discrimination learning task alters tuning instability in the non-CW (surround) compartment. Thus, tuning instability is a robust feature of this compartment that is not predicted by standard models of experience-dependent plasticity, and raises important questions about information processing in this compartment.

Thus, the major advance, which was not known from prior anesthetized or non-longitudinal imaging, is that a specific subset of L2/3 PYR cells – the non-CW tuned cells that are located outside of a whisker’s home column – represent a functionally distinct compartment with highly unstable tuning.

2. It is unclear what the authors mean by “active construction” of untuned neurons; passive receipt of sensory information and Hebbian mechanisms, given the interconnectivity of L2/3 neurons across barrel columns, might be sufficient to generate this effect. In addition, prior studies have established that nearby neurons are more likely to share connectivity; the significance of the 20 um distance for similarly-tuned neurons does not seem surprising. Thus, the conclusion that this instability is highly-structured seems overstated. In addition, the idea that some untuned cells serve to “actively explore novel sensory codes” seems to be unsupported by the data. What is the evidence that they are exploring a code? What code are they exploring – multiwhisker activation?

By ‘active construction’ we only meant that the salt-and-pepper map is not simply inherited from L4, but appears de novo in L2/3, suggesting that it is constructed by cortical circuits. We make this clearer now (Introduction ¶ 1) and remove the word ‘active’ to avoid confusion. We agree with the reviewer that cross-columnar circuits are strong candidates for driving non-CW responses, and the intermixing of cross-columnar synapses, local recurrent synapses, and feedforward synapses onto L2/3 PYR neurons could explain the structure of tuning instability that we observe (see Discussion). But the fact that a plausible circuit basis exists that could explain the observed tuning instability does not mean that such instability is trivially predicted or uninteresting.

We believe it is fair to call tuning instability ‘structured’ in the L2/3 map. This is because significant changes in best whisker (BW) are 3-fold more likely among non-CW tuned cells than CW-tuned cells in the one-vs-all task (Fig. 6a). And in the new, D-vs-C/E discrimination task, BW changes are >6-fold more likely among non-CW tuned cells than CW-tuned cells, where they are essentially absent (Fig. 8g). **Thus, our data clearly show that instability is structured in L2/3, indeed this is the main finding of the paper.** Such structure is important, because it has not been previously observed in other cortical areas, and it rules out the alternative hypothesis that tuning changes are not real, but instead reflect uncontrolled changes in behavioral state or physical stimulus presentation. This alternative hypothesis has been argued to explain away tuning instability in hippocampal place cells⁷. Thus it is appropriate to stress the structured organization of tuning instability in S1, though we removed the one instance in which we had called it ‘highly structured’.

In the original manuscript, we proposed that tuning instability might act to sample possible sensory codes, some of which may be useful and later stabilized. We tested this hypothesis for the revision, by training mice on the D-vs-C/E discrimination task, in which D whisker stimuli are associated

with reward. We found that in D-vs-C/E expert mice, tuning was more stable, on average (Fig. 8f), but that this effect was due to stabilization of CW-tuned cells, while non-CW tuned cells continued to exhibit unstable tuning (Fig. 8g). Thus, tuning instability may be related to learning for CW-tuned cells, but its function among non-CW cells is unclear. We have therefore removed the proposal that instability acts to sample potential sensory codes (see Discussion).

3A. The abstract talks about expert mice, but there is no mention of training, or what the task actually is. The authors should make it clear that neural activity maps were characterized after extensive training in a whisker detection task in both the abstract and introduction. In Figure 1, is day 1 the first day of training? This is not clear from the figure legend, and indeed it is surprising if the mice performed at 80% from the first training day. Secondly, the mice appear to improve their accuracy over the imaging period as shown in Figure 1b. If this is the case, then it is less surprising that surround-tuned neurons adjust their preferred whisker across the imaging period, as behavior is still changing.

Thanks for this comment. We are now much more explicit about task design, and we show a more extended trajectory of behavioral learning in Fig. 1b (previously, Fig. 1b only showed the final 8 days before imaging began). Fig. 1b combines both viral GCaMP6s mice used for acute imaging and transgenic TIGRE2-GCaMP6s mice used for longitudinal imaging; it shows only the first 4 imaging sessions because some viral GCaMP6s mice were only used for a small number of sessions. For the longitudinal imaging mice, we agree that it is essential to show behavioral stability over the whole time course of longitudinal imaging, and this is now shown in Fig. 5b. Thus, tuning instability occurred despite stable mouse performance. We have rewritten the text to be very clear that all imaging occurred in highly trained expert mice.

3B. Does the structure of the task (reward of multiwhisker stimulation) influence the characterized responses – perhaps if just two whiskers were stimulated, the representation of the second stimulated whisker would be stabilized? Could the use of a multiwhisker stimulus coupled to a reward drive the dynamic tuning of the neurons observed?

This is a great question, and we have performed a new experiment to address it (new Figure 8). Our prior task required the mouse to discriminate single-whisker stimuli (unrewarded) from an all-whisker stimulus (rewarded). We call this the one-vs-all discrimination task. In the new experiment, we trained 2 mice on a D-versus-C/E whisker discrimination task in which individual D row whiskers are paired with reward, and C and E-row whiskers are unrewarded. Mice learned over several weeks to lick to D-row whiskers, but not C or E-row whiskers. Thus, this task both associates D-row whiskers with reward, and requires single-whisker discrimination, which our prior one-vs-all task did not.

We performed longitudinal imaging in D-vs-C/E expert mice, and found that overall tuning stability was increased, particularly for cells in D columns, relative to the one-vs-all task. Thus, the behavioral need to discriminate individual whiskers (and/or the association of single-whisker stimuli with reward) stabilizes tuning. But surprisingly, this increase was specific to CW-tuned cells—non-CW tuned cells exhibited the same rate of tuning instability as in our standard one-vs-all expert mice (Fig. 8g). Thus, tuning instability is not simply a product of lack of reward to single-whisker stimuli, but is a property of the non-CW compartment. This is consistent with our one-vs-all expert results, where both CW and non-CW whiskers are nonrewarded, but the non-CW compartment shows preferential tuning instability.

Minor

1. Emx1 mice show abnormal activity in Ca imaging experiments as well as altered dendritic structure and synapse organization. Were there differences in results from Emx and DRd3 animals? Also, what is the background of the Drd3 mice?

We quantitatively compared map structure in Emx1-Cre and Drd3-Cre mice in Suppl. Fig 4. Single-cell whisker tuning and responsiveness were indistinguishable, and map topography differed only slightly between these strains, with somewhat higher local tuning homogeneity in Emx1-Cre mice. We used Emx1-Cre mice for viral expression of GCaMP6s, and we used Drd3-Cre mice for transgenic expression of GCaMP6s, by crossing with the Ai162 (TIGRE2-GCaMP6s) reporter line. We did not use Emx1-Cre to drive transgenic expression of GCaMP6s, because this has been associated with epilepsy⁸. Drd3-Cre mice were originally obtained on a mixed background from MMRC, and were backcrossed to C57BL/6 background for > 8 generations before use in the current study.

2. Was there any direction selectivity to the tuned and less-tuned neurons? Note that L2/3 in adult mice is exactly where directional maps (that show some local clustering) have been described. How deep was the imaging? Was there a difference between deeper and more superficial layers? One might expect that deep L3 would be better tuned than superficial L2.

Unfortunately, we did not measure direction selectivity—all whisker stimuli were caudal-rostral ramp-hold-return stimuli. Imaging depth spanned from 150 to 275 μm below the dura. We observed a trend for deeper cells to have narrower tuning and a more precise topographic map, but this was not statistically significant.

3. What is the rationale for grouping all cells together, and not averaging across animals? What efforts were taken to ensure that there weren't more cells from one animal dominating the average? Do all animals show the same thing?

Cell numbers were approximately equal per mouse (e.g., for longitudinal imaging in one-vs-all task experts, mouse 1-4 yielded 901, 831, 879, and 826 cells for session 1). To show that the major effects were consistent across mice, we now show individual mouse data for i) tuning curve shape and sharpness (Suppl. Fig. 1g-h); ii) tuning heterogeneity within a whisker column (Suppl. Fig. 3b-c), and iii) localization of tuning instability to non-CW tuned cells (Suppl. Fig. 8e).

4. Wouldn't it be more accurate to say that tuning was more stable in CW, rather that instability was concentrated in non-CW neurons?

We believe these are equivalent statements.

5. "... while non-CW neurons may be preferentially read out for coding of non-spatial features." What are the non-spatial features that might be encoded? Is this velocity or frequency? Can the authors be more precise about new experiments and predictions? This would help underscore the significance of the major findings, which otherwise seem very consistent with prior work.

We have updated this section, which now states: "We hypothesize that the core compartment may preferentially carry single-whisker information for touch localization, while the surround compartment may carry more integrative or context-dependent information including for multi-whisker features⁹⁻¹¹."

Reviewer #2

We thank the reviewer for the recognition of the importance of our results.

0. My only comments are on exposition. In general the paper is rather terse; the authors should use a few more words to explain what is going on and what it might mean. This, in addition to a fuller discussion of related literature (including theory) could improve it further by making the paper more accessible and connecting the findings to a wider and deeper literature on this topic.

We have rewritten the text to more clearly introduce each topic, and to connect the findings to broader

conclusions and predictions. We have added fuller discussion of literature and theory, particularly for tuning instability (see Introduction and Discussion).

1. The role of auditory cue should be better motivated from the start. Why is it being used in a tactile task? Is the multisensory part particularly relevant to the investigation? It doesn't seem to be, based on the main findings and conclusions.

The text now explains the behavioral tasks in more detail. In the one-vs-all whisker task (Figs. 1-3, formerly called the 'whisker-cued task'), the auditory stimuli were included in order to be able to compare false-alarm licking to non-rewarded whisker stimuli (single whisker S- stimuli) vs. non-rewarded auditory stimuli (tone S- stimuli). The finding that mice licked more to whisker S- than tone S- stimuli (Fig. 1b-c) indicates that mice in this task are indeed whisker attentive, rather than simply licking to any sensory stimulus. Thus, we know we are studying whisker coding under whisker-attentive conditions. The text now states:

In expert mice, false alarm licking was more common to single-whisker S- than sound S- or blanks, indicating that mice attended whisker stimuli more than auditory distractors (Fig. 1b-c). This task design allows single-whisker stimuli to be used to map receptive fields in whisker-attentive mice without lick contamination or a delay period. (Result p. 4)

We also examined S1 coding of whisker stimuli in the sound-cued task (Fig. 4), which uses the identical stimulus set and trial structure, but in which the mouse must lick to one of the two tones. For this task, mice false-alarm more to the other tone than to single-whisker stimuli, indicating that mice are auditory-attentive in this task. The text now states:

Sound-cued mice exhibited higher false-alarm licking to the tone S- than to whisker stimuli, indicating that they attended to auditory stimuli (Fig. 4b-c). (Results, p. 6)

2. In describing the extraction of calcium signals, the phrase 'neuropil contamination' is a little misleading. Really you are saying that collateral projections may carry signals as well and that you are trying to restrict the analysis to the somata that are within the column. This point could be expanded because it raises questions about whether defining 'barrels' according to the location of somata really makes sense in the first place.

Neuropil contamination is a common concern for 2-photon imaging, and refers to bleeding of signal from nearby out-of-focus dendrites or axons into a somatic ROI. The neuropil signal carries the average tuning of these nearby processes. In our study, the main concern is that neuropil contamination would make the tuning of two nearby somata appear more similar than it actually is—a particular danger in testing whether nearby neurons are part of a co-tuned cluster (Fig. 3). To minimize this concern, we used CalmAn to extract ROI signals throughout the study, because this method minimizes neuropil contamination, and we confirmed the existence of tuning clusters in separate experiments using nucleus-targeted GCaMP6s, which is not expressed in dendrites or axons, eliminating neuropil contamination.

We focus on tuning properties of L2/3 PYR cell somata relative to column boundaries, because our goal is to quantify the columnar organization of whisker information in S1. It is the spatiotemporal activity patterns among PYR somata that define population code that is output from L2/3. Whether barrel boundaries are a relevant framework for whisker information is, ultimately, the main question of our study. The standard conceptual model of a salt-and-pepper representation (e.g. for orientation in mouse V1) is that local circuit connectivity, not cortical location, determines how a neuron contributes to sensory coding. But our results show that among L2/3 PYR cells tuned for the same whisker, CW-

tuned and non-CW neurons (which differ by their columnar location) have different tuning stability and other properties. Thus, neuron location relative to column boundaries does impact coding – i.e., cortical location matters.

3. S1 is described as having prominent experience-dependent plasticity - maybe add a sentence to summarize the relevant type of plasticity that is observed?

We now clarify that L2/3 of sensory cortex exhibits robust plasticity in response to patterns of sensory input (Results, p. 8).

4. The S- condition is not defined explicitly – help the readers more here.

S- refers to unrewarded stimuli, while S+ are rewarded stimuli. This is now stated explicitly (Results. p. 4).

5. The sentence introducing noise correlations - maybe parenthesize the phrase "noise correlations" and not "trial-by-trial covariability"? Many of us feel it is a lingering term that should be retired eventually and the alternative term the authors use is far superior!

We agree. But with the addition of the new Fig. 8, we streamlined some findings to prevent the paper from being unwieldy, so we have removed the noise correlation results.

6. Some of the writing in the main part of the results is very terse and might benefit from reordering, e.g. in the text surrounding Fig 5 I started off asking whether chronic imaging was being done in expert mice, only to find it later on in a very busy paragraph. This applies in the previous section too: it might be better to state upfront the goal of the upcoming set of results and whether each set of results is in trained mice, or mice that are learning to discriminate stimuli.

Thanks for pointing this out. We have rewritten the Results section to more clearly explain the goal of each experiment before presenting the design and the results. We also clarify that all imaging was performed in task-expert mice, not during the initial learning.

7. Fig 5b – it is hard to do the visual longitudinal comparison across these images. Would it be possible to overlay consecutive images somehow, or indicate changes with a marker/silhouette?

This panel is now 5d. We made this panel bigger and aligned each image so it is easier to compare across days. Unfortunately, we could not find a way to overlay the session images while keeping the information of whisker tuning in the same figure.

8. 5f - very busy and hard to parse. Bold red creates a weird optical illusion with the overlaid black text. This is one of the main summaries of one of the main findings, so it would pay to spend some time with this figure to see if it can be made more user friendly and easier on the eye. Ditto 7c.

We changed the red to blue, which is easier to read. We added additional labeling and separated the same-day analysis from the cross-day analysis. We think this helps a lot. We did the same for Fig. 7c.

9. In 1e, 5c - check alignment and sizing of subpanels in the figures. They are crowded and hard to look at in places.

In Fig. 1e, the subpanels have different sizes intentionally, because each subpanel contains a different number of trials (we are showing all trials for each stimulus). We have labeled and spaced these to make them easier to read. For Fig. 5d (former 5c), we enlarged this panel to make it easier to read, and expanded the average soma images to make them easier to see. We have double-checked alignment, and have adjusted the traces in each panel of 5c to improve readability.

10. The conclusion of botox experiments is a little strong. They can't rule out experience-driven plasticity, because whiskers can still transduce stimuli passively. And the botox injection and any subsequent adaptation could itself cause reorganisation of S1 responses over extended periods of time. There is no way to rule out this confound in the present study, so the conclusion should be appropriately mitigated.

It was not our intention to claim that Botox eliminates whisker input. We think that Botox is a useful manipulation because it greatly alters whisker input statistics (e.g., passive input generated by head or body motion will be spatiotemporally very different from active input generated by whisking). Because classical use-dependent Hebbian plasticity is driven by the amount and pattern of sensory input, disruption of input patterns may alter tuning instability if that instability reflects classical use-dependent plasticity. We found no change in tuning instability during Botox (Fig. 7c), or at the onset of Botox or late phases of Botox (Supplementary Fig. 8). Thus we conclude that either tuning instability does not require normal patterns of whisker input, or that even minimal, abnormal sensory input is sufficient to drive tuning instability (p. 8). The overall conclusion is given in Results, p. 8:

These results show that tuning instability does not require normal patterns of whisker experience outside the task. They also confirm that tuning instability is not due to day-to-day variation in whisker tone or whisker self-movement during the imaging sessions. Because whisker paralysis reduces whisker input and disrupts sensory correlations, this suggests either that minimal, abnormal sensory input is sufficient to drive tuning instability, or that an internal source, rather than external sensory input, drives tuning instability.

The amount and prevalence of tuning instability among non-CW cells was unchanged by Botox, and unchanged by associating specific whiskers with the reward in the D-vs-C/E whisker discrimination task (new Figure 8). Thus, tuning instability in the noncolumnar surround is robust to sensory input statistics, whisker-reward association, and the behavioral need to discriminate individual whiskers or not. In contrast, the relatively small amount of tuning instability among CW-tuned cells was altered by D-vs-C/E training, such that tuning became more stable in these mice performing whisker discrimination. We conclude that:

Tuning instability in the non-CW compartment was a robust feature in one-vs-all expert mice, Botox-treated mice, and D-vs-C/E expert mice, suggesting it is an inherent property of the L2/3 map. (Results, p. 11)

11. The authors have cited a few important papers on representational drift, but a number of key references are missing. This is not just about having a 'complete list', there are several that posit the role of drift and the possible causes and these are important to contextualise the findings and show their connection with wider theories of how neural circuits maintain overall function. Lütcke, Margolis & Helmchen TINS (2013) is one of the earliest and most prescient reviews on the phenomenon being studied here. Rule, O'Leary & Harvey, Curr Opin in Neurobiology (2019) offers several specific hypotheses for the roles for drift that speak to the results in this manuscript. Similarly, Raman & O'Leary, eLife (2021) shows why plasticity processes that reorganise neural responses (at the synaptic level) are expected to persist in the absence of a learning signal - a prediction germane to the results around Figure 7. Finally there are a few theory papers that do a nice job of explaining how drift might coexist with stable circuit function: Druckmann & Chklovskii, Curr Biol (2012) somewhat anticipate this (they deal with fluctuations on fast timescales, but the high level point is there). Fauth, Wörgötter & Tetzlaff, PLoS Comp Biol (2015) present a model that maintains function despite synaptic turnover. Rule &

O'Leary, bioRxiv (2021) show how a continually reconfiguring population representation can reliably interact with more stable representations in other circuits. This speaks to the idea that whisker barrels contain an anatomically rigid 'input' (e.g. in L4) while allowing a drifting representation in S1 as a whole. Thanks for these suggestions. We have substantially expanded the Introduction and Discussion to more thoroughly introduce representational drift, including the theoretical context, and to provide additional interpretation of our findings. We cite several of the suggested papers. See Introduction (p. 3) and Discussion (p. 12). One important interpretation of our findings may emerge from the theoretical analysis of Rule & O'Leary 2022, who show that the brain can adapt to accurately read a drifting neural code if a stable internal model is also present that can guide Hebbian and homeostatic plasticity. Both CW-tuned and non-CW tuned neurons are known to project downstream to M1, S2 and other targets. Thus, CW-tuned neurons in S1, which remain yoked to topographic L4 input, may provide this stable internal model to downstream areas as an instructive signal to adapt to the drifting sensory code (Discussion, p. 12).

Reviewer #3

These findings are novel, important and illustrate representation and representational drifts in primary sensory areas in awake mice. In particular, the relationship between anatomy and drift (i.e., the dependency of the drift as function of the distance to the center of the barrel). Moreover, the experimental design and the methods are reproducible. Hence, with moderate revision this study should be accepted as it is clearly suitable to a large audience of system neuroscientists.

We thank the reviewer for these positive comments.

Moderate:

1. As I wrote in my earlier assessment of the paper, the authors have used repetitive stimulation of each whisker to evoke sensory response. Train stimulation leads to substantial adaptation and/or facilitation which clearly alters the receptive field profile of cells (adaptation is also known in awake mice, see Musall et al. 2014). This effect may alter the RF profile. For example, a given cell may adapt more to its corresponding whisker and less to an adjacent whisker. In this case the cell may exhibit stronger calcium signal to the adjacent whisker, despite responding less to the first stimulus in the train. Although I don't ask for additional experiments or analysis, this issue should be discussed.

We agree, and now make this issue explicit. But trains are necessary in this experiment for efficient detection of whisker-evoked GCaMP6s signals. We now state:

Tuning measured from brief whisker deflection trains, as used here, may differ from tuning to single deflections. (Results, p. 5)

2. The authors wrote that they did not observe over-expression, but there are a few clearly over-expressing cells in 1D (filled nuclei). Please exclude them from all the analyses, and clearly specify an exclusion criterion, e.g. using the N/C (nuclear/cytosolic) ratio as in Yang et al. 2018 (NC) or rise/decay dynamics.

N/C ratio has been used in *in vitro* imaging from cultured cortical neurons to identify GCaMP overexpression (Yang et al. 2018). What value of N/C ratio identifies unhealthy neurons *in vivo* is not known. To test this, we analyzed data from the final imaging session (i.e. longest duration of expression) from our longitudinal imaging dataset, in six *Drd3-Cre:TIGRE2-GCaMP6s* mice. We analyzed all cells with unimpeded imaging of somatic cytosol (i.e. no overlap with other cells or dendrites) and with a large enough diameter to be confident that the nucleus was contained in the optical plane of section (n=1896 cells). Because a primary sign of overexpression is reduced responsiveness, we first asked whether responsive vs. non-responsive cells differed in N/C ratio. They did not (p=0.269, KS test) (**Fig. R1a**). We

next asked whether CW-tuned vs. non-CW tuned cells differed in N/C ratio, to ensure that non-CW tuning did not represent abnormal responsiveness due to overexpression. There was no difference in N/C ratio ($p=0.258$, KS test) (**Fig. R1b**).

Next, we examined $\Delta F/F$ dynamics. Rise/decay dynamics are influenced both by single-spike Ca^{2+} dynamics and by the temporal structure of sensory-evoked and spontaneous spike trains. We focused on decay kinetics, which are strongly linked to overexpression. We identified 3478 cells in the same imaging sessions as above that exhibited identifiable isolated Ca^{2+} transients, defined as a transient that had no other peaks ($> \text{mean} + 1 \text{ s.d.}$ of blank trials) in the prior or subsequent 2 seconds. The mean decay half-time was $825 \pm 180 \text{ ms}$, which is well within the range of healthy GCaMP6s kinetics reported previously in V1 *in vivo*^{12, 13} (**Fig. R1c**). Finally, we tested whether cells with the highest N/C ratio had slowest DF/F dynamics, as expected if these cells had unhealthy levels of GCaMP6s expression. Of the cells whose N/C ratio was quantifiable (from the prior paragraph), we identified 1887 cells which also had identifiable isolated Ca^{2+} transients. Among these neurons, there was no relationship between N/C ratio and decay half-time (**Fig. R1d**).

Thus, while we cannot rule out that some small fraction of cells had GCaMP6s overexpression, we did not find that N/C ratio predicted response abnormality, tuning abnormality, or abnormal decay kinetics. This results are described in Methods (p. 17).

Fig. R1. Analysis of GCaMP6s overexpression by N/C ratio and $\Delta F/F$ decay kinetics. A, Distribution of N/C ratio values for responsive vs. non-responsive cells. Gray shows all cells. Inset: examples of cells with three different N/C ratios. B, Distribution of N/C ratio for CW-tuned cells ($\text{BW}=\text{CW}$) and non-CW tuned cells ($\text{BW}\neq\text{CW}$). P-values are KS-test. C, Distribution of decay tau for spontaneous isolated DF/F transients. Inset bars show decay tau ranges for single spike DF/F responses in anesthetized V1 (from Ref. 12) and in response to drifting grating stimuli (from Ref. 13). D, Cells with higher N/C ratio do not have longer decay kinetics. P-value is from t-test for non-zero slope.

3. It is important to see some additional analysis of the ‘wavering’ characteristics of some of the cells (i.e., cells that lost their whisker responsiveness completely or gained it from none) with respect to the cells’ locations and their CW status, similar to the analysis that is done with the changes in BW or CoM. This is crucial as the fraction of such cells is even larger than the fraction of cells that were responsive across both sessions. Are wavering cells more likely to be septal or non-CW tuned than CW-tuned? What is the function between the wavering likelihood and the barrel center?

We have added an analysis of ‘wavering’ cells (Fig. 6g-i, and Suppl. Fig. 8g). Unlike cells with tuning instability, wavering cells were located equally at all map locations (Fig. 6g). The strongest predictor of whether a responsive cell would subsequently lose responsiveness was its initial whisker response strength—with weakly responsive cells more likely to waver in their responsiveness over days (Fig. 6h). Correspondingly, when we quantified the weight of various factors in predicting tuning instability or wavering responsiveness, we found that spatial location in the map was the strongest predictor of tuning instability, but initial response strength was the strongest predictor of wavering responsiveness (Suppl. Fig. 8f-g).

This suggests that tuning instability and wavering responsiveness are independent phenomena. Consistent with this idea, we tested whether wavering cells were more or less likely to change their tuning when they became responsive again. The probability of tuning change over 3 sessions was identical for wavering cells vs. consistently responsive cells (Results, p. 7). Thus, tuning instability is distinct from wavering responsiveness, and affects both wavering and non-wavering neurons equally.

4. The authors wrote “GCaMP6s expression was stable throughout”, but this should be supported with a figure or at least a calculation. To show GCaMP expression is stable throughout, plot the magnitude of baseline fluctuations (ongoing activity) for each cell over time, and check if there is an overall trend.

We have added this analysis (Fig. 5c). For each imaging session, we calculated the mean spontaneous event rate and amplitude for $\Delta F/F$ transients in each blank trial. Spontaneous events were identified as those $\Delta F/F$ transients whose peak exceeded the mean plus one s.d. of $\Delta F/F$ on that trial. For each imaging field, the mean event rate and amplitude were calculated across all neurons. Mean event rate was stable over the 4 pre-Botox sessions and the 4 Botox sessions (now reported in Fig. 5c). Spontaneous event magnitude was also largely stable across these sessions, shown here for the reviewer (Fig. R2):

Figure R2. Magnitude of spontaneous $\Delta F/F$ events across 8 sessions in 16 imaging fields (4 mice). Each point is the average magnitude of spontaneous events, calculated across all cells in the imaging field. These are the same events whose mean frequency is reported in Fig. 5c.

5. Can the authors check stability of the representations in the data that comes from the auditory task? It is well possible that drifts are different in animals that trained for the whisker task than naïve mice. Although doing additional experiments can be useful, I believe that looking at the data from the auditory task will be good enough to address this issue.

Unfortunately, the sound-cued experiments were done using viral GCaMP6 expressing mice, not transgenic TIGRE2.0-GCaMP6s, so we do not have longitudinal imaging data for this experiment. However, we have added a new experiment (in response to Reviewer 4 Point 1) in which we tested the fundamental question that Reviewer 3 is asking here – whether association of whisker stimuli with reward, or the nature of the sensory task, impacts tuning stability. These results show that stability is affected by behavioral training – but only for CW-tuned cells, with non-CW tuned cells showing normal high levels of tuning instability. Please see response to Reviewer 4, Point 1.

Minor:

1. In several places there is use of the word ‘identical’, e.g. “Identical local clustering of tuning similarity was observed” and “PYR cell response amplitude and tuning sharpness were identical in SC and WC mice”. Please remove these uses and in each place state that you did not find a statistically significant difference.

We removed “identical” from these statements.

2. You write “unstably and stably tuned neurons showed similar responsiveness and tuning sharpness in the first session” but the proportions look somewhat different. Please quantify as you do with KS or chi-squared for other comparisons.

Here, we were only trying to point out that both unstably and stably tuned neurons showed strong whisker responses and good tuning in the initial imaging session. We have changed the wording to indicate this, and not to claim that these are quantitatively identical (Results, p. 6).

3. More details on the masking noise are needed in the methods section.

We have added details on the structure and sound level of the masking noise relative to other noise sources in the task environment (Methods, p. 14).

Reviewer #4

Overall this is a well performed set of experiments, and I enjoyed reading it – especially the rigour of the experiments and their analysis. My main concern is that although a nice set of experiments done well, it seems very descriptive. My worry is the manuscript does not fully convey the crucial biological implications of these findings, and the consequences of this organisation for circuit function is what would be more appropriate, rather than just the organisation itself. Some specific thoughts associated with this are expanded on below, and could potentially be addressed with a substantial extra discussion, but ideally there would be at least some experimental evidence for what these findings might mean more broadly for the functioning of the circuit.

We address these concerns in our response to the specific points, below. Broadly, we have expanded the paper both experimentally and in discussion of theoretical concepts. In the original manuscript, we made two predictions for how map organization and tuning stability relate to learning: first, that non-CW tuned cells may be the main site of learning-related somatotopic plasticity; and second, that learning may stabilize neural tuning that is beneficial to task performance. We gathered some initial data in support of the first prediction, but realized that a strong test requires development of new, more rapidly learned tasks (see Reviewer 4 Point 1). Because this will take substantial time, we feel that it is more appropriate for a separate study. For the second prediction, we performed a new experiment (Fig. 8) testing whether tuning instability is affected by training on different tasks. The results show that behavioral use indeed modulates tuning instability, stabilizing tuning when individual whiskers need to be discriminated. This provides a concrete link between tuning instability and learning (see Reviewer 4 Point 2).

We also expanded our discussion of the theoretical implications of having intermixed stable and unstable representations in sensory cortex. Prior theoretical work predicts the critical importance of a stable internal sensory model in order to guide adaptive learning of shifting sensory codes (e.g., Rule & O’Leary, 2022¹). Because both CW-tuned and non-CW tuned neurons project to downstream areas, our results suggest that CW-tuned cells provide this stable internal model to downstream areas, while non-CW tuned cells provide the beginnings of an integrative but drifting neural code (see Reviewer 2 Point 11).

Main points:

1. The authors propose that due to their more unstable dynamics, non CW tuned cells may play a preferential role in plasticity, compared to the more stable CW tuned cells acting as a ‘backbone’, but this is not tested (for example by monitoring the activity across learning, or across a contingency change). It feels like all of the experiments in the manuscript are leading up to that point, but then the

experiment to explicitly test this is missing. In my opinion, the manuscript would be substantially strengthened if experimental or computational insight into the differential contribution of these two classes of cells could be provided.

In the original manuscript, we had proposed that non-CW tuned cells are the major site of receptive field plasticity during learning. We performed an initial test of this hypothesis and obtained some promising results (described here for the reviewer). But we realized that a rigorous test will require development of new, more rapidly learned whisker sensory behaviors. Developing such behaviors is slow, and we believe is not tenable within the time frame of the current study.

As an initial test of whether non-CW tuned cells are the major site of receptive field plasticity during learning, we first trained 2 mice on the sound-cued task (in which whisker stimuli are behaviorally irrelevant), and used imaging to identify CW- and non-CW tuned cells. Next, we retrained these same mice on a new task, the D-vs-C/E whisker discrimination task, in which mice learn to lick to D-whisker stimuli (rewarded) but not to C or E whisker stimuli (unrewarded). Training took 3 weeks. Training on this task creates a stronger, more columnar representation of the D whiskers in L2/3 (described in Reviewer 4, Point 2). We followed individual neurons longitudinally across these two tasks. We quantified the probability that a neuron that was originally tuned for a C or E whisker switched its best whisker to become tuned for a D-whisker after D-vs-C/E training. Non-CW neurons showed a high probability of switching to D tuning (0.30), while CW neurons had a negligible probability of switching (0.06, where chance is 0.05). These data are based on 114 cells that were originally tuned to C/E whiskers (6 imaging fields, 2 mice). These results are consistent with the hypothesis that non-CW cells are the site of most learning-related tuning changes.

However, it was apparent from this experiment that to study this well, we need to use behavioral training paradigms with faster learning, because the long duration of learning reduces longitudinal imaging yields, and makes it hard to isolate learning-specific changes in whisker tuning. In addition, to be compelling, we would need several distinct learning paradigms, to test whether non-CW cells are consistently the preferential site of plasticity. This is a big undertaking that is outside the time scale of the current study.

Thus, for the current paper, we have shifted our focus away from how tuning instability relates to use-dependent receptive field plasticity. Instead, we focus on whether behavioral use modulates tuning instability, by asking whether training mice to discriminate single whiskers (using the D-vs-C/E discrimination task) stabilizes tuning compared to mice that are expert at the one-vs-all task, which does not require whisker discrimination. The answer is yes, and this is now Figure 8 (see Reviewer 4, Point 2).

2. The explanation of the relationship between the experiments and the task being performed was a bit vague, and could be laid out more clearly. To someone outside the project it is not immediately clear to what extent the findings are dependent on the specifics of the task – why is the task designed as it is, and how does this specifically help the analysis? More generally, as the authors are explicitly comparing their findings to anaesthetised recordings looking at passive whisker tuning, it would be good to explain the decision to investigate this in context of a learnt task? Is it simply to keep the mouse ‘whisker attentive’? Could the specifics of the learnt task be altering the organisation and function of the circuit? This is addressed for the distribution finding with the sound task in Fig 4, but not for the stability of tuning, which is the main take home and potentially most related to behavioural strategy.

Related to above, from a learning point of view, the design of the task seems initially a bit lopsided, as in essence the mouse is being asked to identify ‘all whiskers’ vs ‘one whisker’ which could be interpreted as a threshold or summation task rather than a discrimination. It would be very beneficial to explicitly explain why this design was chosen – presumably to allow receptive field mapping of each whisker during CS- so as to control for reward artifacts? However in addition, it would be beneficial to have critical discussion of the alternative possibilities or specificity of the findings to the task. For

example, could the organisation found be due to the specific demands of the task? In this specific task the discrimination depends on the ability to integrate stim from all whiskers (CS+) and compare to input from just one whisker (CS-) Could this contribute to the instability in 'outside column' neurons? As in the task the identity of the individual whisker is irrelevant to the learnt task, all that is needed to be known is if it is stimulated with others.

We have made substantial additions to address these questions. We explain the rationale behind the task design more clearly. The reviewer is correct that our main task is a one-vs-all discrimination task, which does not require discrimination of individual whiskers, and is likely to be solved as a global intensity discrimination task. We now describe the goals of the task design in more detail, and name it the "one-vs-all" task:

We developed a novel head-fixed whisker discrimination task to allow measurement of receptive fields in awake, whisker-attentive mice (Fig. 1a)... In expert mice, false alarm licking was more common to single-whisker S- than sound S- or blanks, indicating that mice attended whisker stimuli more than auditory distractors (Fig. 1b-c). This task design allows single-whisker stimuli to be used to map receptive fields in whisker-attentive mice without lick contamination or a delay period... Thus, this is a passive one-versus-all whisker discrimination task. (Introduction, p. 3).

We tested whether tuning instability was related to the structure of this task, in a new experiment (**new Fig. 8**). The one-vs-all task does not require discrimination between individual whiskers, so may not require stable representation of single whiskers in L2/3. The broader question is whether behavioral use of whisker input (single-whisker discrimination vs. global intensity) or training on a specific whisker task modulates tuning instability. To test this, we trained new mice on a whisker identity discrimination task in which they had to lick to single-whisker deflection of any D-row whisker (rewarded), but not to deflection of any single C- or E-row whisker (unrewarded). A delay period separated sensory from lick-related activity in S1. Once mice were experts, we performed longitudinal imaging to assess tuning stability. We found that D-vs-C/E discrimination training increased the fraction of CW-tuned neurons in D columns, and strengthened and sharpened whisker tuning for these cells, indicating that it created a stronger, more columnar map of D whiskers in L2/3. Tuning instability was also sharply reduced (Fig. 8f). Strikingly, this increase in stability was observed only for CW-tuned neurons, and was unchanged for non-CW tuned neurons relative to one-vs-all expert mice (Fig. 8g).

Thus, tuning instability is indeed modulated by learning and behavioral use, but only for CW-tuned neurons. The net effect is to create a sharper, more stable columnar representation in mice that need to discriminate individual whiskers. Among non-CW tuned neurons, tuning instability was identical in one-vs-all trained mice, mice with Botox whisker paralysis, and C-vs-D/E trained mice, indicating that it is a robust feature of this map compartment.

We summarize these conclusions in the Discussion (p. 11):

Second, tuning instability could relate to behavioral demands for sensory processing. In particular, the one-versus-all task does not require single-whisker discrimination, so somatotopic tuning stability may be unnecessary for behavior. To test this, we examined tuning instability in expert mice on the D-vs-C/E task, which requires single whisker discrimination. In D-vs-C/E expert mice, L2/3 cells in D columns exhibited more tuning homogeneity for the CW, and modestly stronger and sharper tuning for D-row whiskers, consistent with a stronger, more precise columnar representation of D-row whiskers. Tuning instability was sharply reduced relative to one-versus-all mice, both broadly across all cells in all columns and even more strongly for CW-tuned cells in D columns. We were able to

longitudinally track a relatively small number of CW-tuned cells in C and E columns, which showed a similar trend toward increased stability (data not shown), suggesting that stabilization is related to behavioral discrimination, rather than specific whisker-reward association. Surprisingly, increased stability was localized not to non-CW neurons, but to CW-tuned neurons, where it reduced instability substantially (Fig. 8h). As a result, in D-vs-C/E expert mice, tuning instability was even more localized to non-CW neurons than in one-vs-all mice.

This stabilization of single-whisker tuning for CW-tuned neurons predicts more stable population coding of whisker identity in L2/3, beneficial for task performance. Overall, these findings are consistent with a model in which training causes some non-CW tuned neurons to shift their tuning to become CW-tuned, where tuning stabilizes. These results show that the prevalence of tuning instability in CW-tuned cells is related to behavioral demands. In contrast, the origin and function of tuning instability among non-CW cells remain unclear.

Technical point:

3. I think it is important to have some more explanation of why *drd3* and *emx cre* were used, and the potential caveats that might arise from imaging subpopulations of neurons when considering the distribution of cells in the circuit. This discussion could include a comparison to e.g. viral nuclear *gcamp* utilised in figure 3. Overall I think the fact that similar findings are shown across multiple expression strategies is a good thing – but it is worth discussing fully and openly.

We initially characterized map organization using *Emx1-Cre* mice with viral *GCaMP* expression. For longitudinal imaging, we switched to *Drd3-Cre:TIGRE2.0-GCaMP* mice, which provide more stable long-term expression, uniform *GCaMP* expression across columns, and lower neuropil signal (because L4 and L5 neurons do not express *GCaMP*). We compare receptive field and map properties between *Emx1-Cre/viral* and *Drd3-Cre:TIGRE2.0-GCaMP* mice in Suppl. Fig. 4, finding only modest differences. We did not use *Emx1-Cre* driven transgenic expression of *GCaMP*, because these have been associated with epilepsy⁸. This is now explained in Methods (p. 14, “Animals”).

Minor:

4. A personal preference, but there are a huge number of acronyms – I had to keep looking back to work out what was being talked about – I would recommend just spelling it out each time.

To reduce the acronyms, we eliminated WC and SC from the text, spelling out whisker-cued and sound-cued instead. We feel that CW, non-CW, BW and CoM (defining specific whiskers and tuning properties) are essential and are used so often that they need to be abbreviated, so we have retained these.

5. It would be very beneficial – especially as this is a new task – to present the data for learning of the task as well as just the days before expts started – how long did it take to get to criterion, and did learning rate correlate with measures studied?

We now show the learning curve for the one-vs-all task in Fig. 1b, and state the number of days to criterion in the Results (p. 4). Because training occurred in stages (learn to lick for reward; learn to lick to S+ stimuli with a small number of S- distractors; then add the full set of S- stimuli), it is difficult to quantify the entire training process, so we show the learning curve for the last stage as the full set of S- stimuli are added. We provide a detailed explanation of the training stages in Methods (p. 15-16). We feel the mouse population size (N=4 for longitudinal imaging on the one-vs-all task) is too low to meaningfully correlate learning rate with tuning instability or other properties.

Citations

1. Rule, M.E. & O'Leary, T. Self-healing codes: How stable neural populations can track continually reconfiguring neural representations. *Proc Natl Acad Sci U S A* **119** (2022).
2. Clancy, K.B., Schnepel, P., Rao, A.T. & Feldman, D.E. Structure of a single whisker representation in layer 2 of mouse somatosensory cortex. *J Neurosci* **35**, 3946-3958 (2015).
3. LeMessurier, A.M., *et al.* Enrichment drives emergence of functional columns and improves sensory coding in the whisker map in L2/3 of mouse S1. *Elife* **8** (2019).
4. Sato, T.R., Gray, N.W., Mainen, Z.F. & Svoboda, K. The functional microarchitecture of the mouse barrel cortex. *PLoS Biol* **5**, e189 (2007).
5. Lyall, E.H., *et al.* Synthesis of a comprehensive population code for contextual features in the awake sensory cortex. *Elife* **10** (2021).
6. Kremer, Y., Léger, J.F., Goodman, D., Brette, R. & Bourdieu, L. Late emergence of the vibrissa direction selectivity map in the rat barrel cortex. *J Neurosci* **31**, 10689-10700 (2011).
7. Liberti, W.A., 3rd, Schmid, T.A., Forli, A., Snyder, M. & Yartsev, M.M. A stable hippocampal code in freely flying bats. *Nature* **604**, 98-103 (2022).
8. Steinmetz, N.A., *et al.* Aberrant Cortical Activity in Multiple GCaMP6-Expressing Transgenic Mouse Lines. *eNeuro* **4** (2017).
9. Laboy-Juárez, K.J., Langberg, T., Ahn, S. & Feldman, D.E. Elementary motion sequence detectors in whisker somatosensory cortex. *Nat Neurosci* **22**, 1438-1449 (2019).
10. Estebanez, L., Bertherat, J., Shulz, D.E., Bourdieu, L. & Léger, J.F. A radial map of multi-whisker correlation selectivity in the rat barrel cortex. *Nat Commun* **7**, 13528 (2016).
11. Vilarchao, M.E., Estebanez, L., Shulz, D.E. & Férézou, I. Supra-barrel Distribution of Directional Tuning for Global Motion in the Mouse Somatosensory Cortex. *Cell Rep* **22**, 3534-3547 (2018).
12. Chen, T.W., *et al.* Ultrasensitive fluorescent proteins for imaging neuronal activity. *Nature* **499**, 295-300 (2013).
13. Lur, G., Vinck, M.A., Tang, L., Cardin, J.A. & Higley, M.J. Projection-Specific Visual Feature Encoding by Layer 5 Cortical Subnetworks. *Cell Rep* **14**, 2538-2545 (2016).

REVIEWER COMMENTS

Reviewer #4 (Remarks to the Author):

I thank the authors for taking the time to do a detailed update of their manuscript, which has addressed my concerns and I now think will make a very nice contribution to the field.

Reviewer #5 (Remarks to the Author):

Han Chin Wang laboratory investigated mouse barrel cortex “salt-and-pepper” maps of sensory features in layer 3 cells in awake mice performing one-vs-all whisker discrimination. They report that neurons tuned for columnar (CW) and non-columnar (non-CW) whiskers are spatially intermixed, with co-tuned neurons forming novel local clusters. “Surprisingly”, whisker tuning was markedly unstable in “expert , trained” mice, with 35-46% of pyramidal cells significantly shifting tuning over 5-18 days. Tuning instability is concentrated in non-CW tuned neurons, and thus was structured in the map. Instability of non-CW neurons was unchanged during chronic whisker paralysis and when mice discriminated individual whiskers, suggesting that it is an inherent feature. They conclude that, L2/3 combines two distinct, “previously unrecognized” components: a stable columnar framework of CW-tuned cells that may promote spatial perceptual stability, plus an intermixed, non-columnar surround with highly unstable tuning.

Unfortunately, this study and the results reported therein begs the question “So what?” It may be of interest to scientists who are studying the minutia of the barrel cortex of mice but it does not reveal any fundamental mechanisms or insights to the mammalian cortical organization and function in general.

1. It is an interesting and curious historical trend that whisker-barrel research in mice has closely followed/copied the footsteps of the mammalian visual system research. There are numerous publications on the “salt and pepper” functional responses from the visual cortex (PMID 24492085, 32160536, 33849948, 17720489) and also a few from the rodent barrel cortex that followed them (e.g., PMID 28918260, 30618647).

2. While this study seems to be carefully conducted and Wang et al., attempted to answer the reviewer concerns with new experiments (unfortunately, by hand-waving), the significance of this study eludes the reader.

3. I concur with the Reviewer 1 that “The main advance appears to be the analysis of maps in awake animals performing a task. However, the results appear largely similar to findings from anaesthetized mice. It has been well-established that L2/3 neurons show variable tuning, and that altered sensory experience can shift tuning preferences. The novelty of the work is reduced by the large body of prior studies, both cited and uncited, that show a similar mapping. Although the analysis of tuning stability, using the power of repeat imaging, is more novel, these results are mainly descriptive and their significance or how they revise current models is not well-developed.”

The authors response to that important criticism is: “ The major findings are the new features of map organization that we discovered through longitudinal imaging. These include the high prevalence of tuning instability (Fig. 5), the fact that tuning instability is structured within the map (Fig. 6), and the fact that instability in the non-CW surround is highly robust to sensory experience and learning manipulations (Figs. 7-8). Together, these findings demonstrate that there are novel, functionally distinct compartments within the L2/3 map, shown schematically in Fig. 6j. This transforms our understanding of the L2/3 map from simply a topographically imprecise map of intermixed tuning ensembles (similar to orientation or retinotopy in mouse V1) to a map in which each whisker is represented by a stable core of neurons with strong columnar topography, surrounded by a broad, non-columnar surround of neurons whose tuning dances unpredictably over days. We propose distinct functions for these novel columnar and non-columnar compartments. This is a major revision in understanding S1 organization”.

4. The authors’ longitudinal imaging is deceptive because they are not imaging across important developmental time points of whisker-barrel development in mice. Their “longitudinal imaging” is

operationally defined for a few time points in adult laboratory mice. They overlook the fact that there are mouse strains that are born with whiskers and whisk shortly thereafter (e.g., African spiny mice), and there are very early born marsupials (e.g., kangaroos and wallabies) that have whiskers and barrels while in their mothers' pouch. I recommend that the authors expand their horizons beyond laboratory mice and rats, the whisker-barrel world is amazingly diverse and murinea is not always a good reference point in understanding the organization and function of the mammalian and human brain. But if their interest is limited to understanding mouse whisker-barrel system, then they should submit this paper to a specialty journal appealing to that audience.

5. Is what they study, and their results, specific to *mus musculus* (or perhaps *rattus Norvegicus*) or do they have any bearing to the organization and function of mammalian cortex? For example, there are several species with "whiskers" that do not whisk and/or have no cortical barrels (squirrels, cats, dogs, seals etc.) There are also species with whiskers that have barrels only in development but as they grow up they "lose them" (e.g., rabbits, marsupials).

6. I fully support neuroscience research on barrel cortex in mice and rats, as these studies have provided valuable information on lamination of cortex, cell migration, axon guidance, map formation, cortical allocations, corticocortical and callosal connections, map formation, plasticity, and so on. But, this study provides hardly any new information on the principles, organization, and function of the mammalian cortex, it is more like a curiosity study that is perhaps discussed at an SfN satellite meetings on whiskers and barrels or more suitable for specialty journals for whisker research, like Somatosensory Motor Investigations (? , not sure of the exact name). I am not undermining the strength of experimental approach and data analysis and the effort, Wang laboratory has put into this paper. However, these findings are far too specific to be of interest to general scientific (and neuroscience) audience. They do not provide any new insights into the organization and function of the mammalian neocortex, or to neuroscience in general.

7. A note to the authors: African spiny mouse start whisking as soon as they are born and have cortical barrels. Even when their whisker pads are ablated in adulthood, they generate new whisker pads and barrels. So the findings reported here ("inherent features of L2/3 cells that combine two distinct, "previously unrecognized" components: a stable columnar framework of CW-tuned cells that may promote spatial perceptual stability, plus an intermixed, non-columnar surround"...) may not even have a universal validity in the murinae. Something to think about.

Response to Reviewers

We thank the reviewers and editor for their comments. We have addressed them by edits to the manuscript. We have also substantially shortened the text to fit in the length requirements.

REVIEWERS' COMMENTS

Reviewer #4 (Remarks to the Author):

I thank the authors for taking the time to do a detailed update of their manuscript, which has addressed my concerns and I now think will make a very nice contribution to the field.

We thank the reviewer for their comments.

Reviewer #5 (Remarks to the Author):

Han Chin Wang laboratory investigated mouse barrel cortex “salt-and-pepper” maps of sensory features in layer 3 cells in awake mice performing one-vs-all whisker discrimination. They report that neurons tuned for columnar (CW) and non-columnar (non-CW) whiskers are spatially intermixed, with co-tuned neurons forming novel local clusters. “Surprisingly”, whisker tuning was markedly unstable in “expert , trained” mice, with 35-46% of pyramidal cells significantly shifting tuning over 5-18 days. Tuning instability is concentrated in non-CW tuned neurons, and thus was structured in the map. Instability of non-CW neurons was unchanged during chronic whisker paralysis and when mice discriminated individual whiskers, suggesting that it is an inherent feature. They conclude that, L2/3 combines two distinct, “previously unrecognized” components: a stable columnar framework of CW-tuned cells that may promote spatial perceptual stability, plus an intermixed, non-columnar surround with highly unstable tuning.

Unfortunately, this study and the results reported therein begs the question “So what?” It may be of interest to scientists who are studying the minutia of the barrel cortex of mice but it does not reveal any fundamental mechanisms or insights to the mammalian cortical organization and function in general.

Our findings are relevant to salt-and-pepper maps in general, by revealing that these maps can contain multiple functionally distinct compartments, and that the topographic component of these maps may be a stable backbone around which an unstable, non-topographic component is organized.

1. It is an interesting and curious historical trend that whisker-barrel research in mice has closely followed/copied the footsteps of the mammalian visual system research. There are numerous publications on the “salt and pepper” functional responses from the visual cortex (PMID 24492085, 32160536, 33849948, 17720489) and also a few from the rodent barrel cortex that followed them (e.g., PMID 28918260, 30618647).

Yes, we are building on those discoveries to document new features of salt-and-pepper maps.

2. While this study seems to be carefully conducted and Wang et al., attempted to answer the reviewer concerns with new experiments (unfortunately, by hand-waving), the significance of this study eludes the reader.

Without specifics, it is difficult to address this criticism. We discuss significance in the Introduction and Discussion.

3. I concur with the Reviewer 1 that “The main advance appears to be the analysis of maps in awake

animals performing a task. However, the results appear largely similar to findings from anaesthetized mice. It has been well-established that L2/3 neurons show variable tuning, and that altered sensory experience can shift tuning preferences. The novelty of the work is reduced by the large body of prior studies, both cited and uncited, that show a similar mapping. Although the analysis of tuning stability, using the power of repeat imaging, is more novel, these results are mainly descriptive and their significance or how they revise current models is not well-developed."

The authors response to that important criticism is: " The major findings are the new features of map organization that we discovered through longitudinal imaging. These include the high prevalence of tuning instability (Fig. 5), the fact that tuning instability is structured within the map (Fig. 6), and the fact that instability in the non-CW surround is highly robust to sensory experience and learning manipulations (Figs. 7-8). Together, these findings demonstrate that there are novel, functionally distinct compartments within the L2/3 map, shown schematically in Fig. 6j. This transforms our understanding of the L2/3 map from simply a topographically imprecise map of intermixed tuning ensembles (similar to orientation or retinotopy in mouse V1) to a map in which each whisker is represented by a stable core of neurons with strong columnar topography, surrounded by a broad, non-columnar surround of neurons whose tuning dances unpredictably over days. We propose distinct functions for these novel columnar and non-columnar compartments. This is a major revision in understanding S1 organization".

The basic salt-and-pepper organization is indeed similar to anesthetized mice. The novelty here is in the discovery of tuning instability and its specific structure in the map. This demonstrates that two functionally distinct compartments exist in the salt-and-pepper map, which was not known previously. The criticism that "it has been well-established that ... altered sensory experience can shift tuning preferences" is off-base: the whole point of our study is that tuning preferences shift substantially while task performance remains stable. This is the central puzzle of tuning instability and representational drift as studied in other cortical areas. Prior studies had concluded that primary sensory cortex does not show substantial tuning instability for local sensory feature tuning. We show that is wrong -- there is marked instability even in primary sensory cortex.

4. The authors' longitudinal imaging is deceptive because they are not imaging across important developmental time points of whisker-barrel development in mice. Their "longitudinal imaging" is operationally defined for a few time points in adult laboratory mice. They overlook the fact that there are mouse strains that are born with whiskers and whisk shortly thereafter (e.g., African spiny mice), and there are very early born marsupials (e.g., kangaroos and wallabies) that have whiskers and barrels while in their mothers' pouch. I recommend that the authors expand their horizons beyond laboratory mice and rats, the whisker-barrel world is amazingly diverse and murinea is not always a good reference point in understanding the organization and function of the mammalian and human brain. But if their interest is limited to understanding mouse whisker-barrel system, then they should submit this paper to a specialty journal appealing to that audience.

Our study is not about development. It is about understanding stability and instability in population coding in adults under stable behavioral conditions. This is the intellectual question being addressed in the 'representational drift' field at large. The introduction now explicitly states that this is a question about adult brain function (lines 19-20).

5. Is what they study, and their results, specific to *mus musculus* (or perhaps *rattus Norvegicus*) or do they have any bearing to the organization and function of mammalian cortex? For example, there are several species with "whiskers" that do not whisk and/or have no cortical barrels (squirrels, cats, dogs, seals etc.) There are also species with whiskers that have barrels only in development but as they grow up they "lose them" (e.g., rabbits, marsupials).

It is well-known that only some species have whisker barrels. Basic cortical circuits are highly similar between mouse barrel cortex and visual cortex, suggesting that these are both useful model systems for understanding principles of sensory cortex function. The presence of anatomical column boundaries is technically useful for our experiments, but our results are relevant for non-

anatomically-columnar areas as well. For example, the tonotopic map in L2/3 of mouse auditory cortex has a very similar salt-and-pepper topography, superimposed on the mean tonotopic map, as does whisker S1. Our results suggest that a similar core-and-surround structure may exist with distinct levels of tuning instability in these compartments, in auditory cortex (and in visual cortex) as well as in S1. We added a sentence to the discussion to highlight this:

Other salt-and-pepper maps, such as for tonotopy in auditory cortex^{54, 55} and retinotopy in visual cortex⁵⁶, may also exhibit distinct stable vs. more integrative, unstable compartments. [p. 11].

6. I fully support neuroscience research on barrel cortex in mice and rats, as these studies have provided valuable information on lamination of cortex, cell migration, axon guidance, map formation, cortical allocations, corticocortical and callosal connections, map formation, plasticity, and so on. But, this study provides hardly any new information on the principles, organization, and function of the mammalian cortex, it is more like a curiosity study that is perhaps discussed at an SfN satellite meetings on whiskers and barrels or more suitable for specialty journals for whisker research, like Somatosensory Motor Investigations (? , not sure of the exact name). I am not undermining the strength of experimental approach and data analysis and the effort, Wang laboratory has put into this paper. However, these findings are far too specific to be of interest to general scientific (and neuroscience) audience. They do not provide any new insights into the organization and function of the mammalian neocortex, or to neuroscience in general.

We obviously disagree, and can cite many papers on circuit function, population coding, and principles of L2/3 function that were performed in S1 and are published in top-tier, non-specialty journals.

7. A note to the authors: African spiny mouse start whisking as soon as they are born and have cortical barrels. Even when their whisker pads are ablated in adulthood, they generate new whisker pads and barrels. So the findings reported here (“inherent features of L2/3 cells that combine two distinct, “previously unrecognized” components: a stable columnar framework of CW-tuned cells that may promote spatial perceptual stability, plus an intermixed, non-columnar surround”...) may not even have a universal validity in the murinae. Something to think about.

This sounds like a fascinating species. We would fully expect that during injury-dependent remapping and regeneration, like during learning, receptive fields of many neurons would shift substantially.

Sincerely,

Dan Feldman & Han Chin Wang